# 2D transition metal dichalcogenides with glucan multivalency for antibody-free pathogen recognition

Tae Woog Kang[1], Juhee Han[1], Sin Lee [1], In-Jun Hwang[1], Su-Ji Jeon[1], Jong-Min Ju[1], Man-Jin Kim[1], Jin-Kyoung Yang[1], Byoengsun Jun[2], Chi Ho Lee[2], Sang Uck Lee [2] & Jong-Ho Kim[1]

The ability to control the dimensions and properties of nanomaterials is fundamental to the creation of new functions and improvement of their performances in the applications of interest. Herein, we report a strategy based on glucan multivalent interactions for the simultaneous exfoliation and functionalization of two-dimensional transition metal dichalcogenides (TMDs) in an aqueous solution. The multivalent hydrogen bonding of dextran with bulk TMDs ($WS_2$, $WSe_2$, and $MoSe_2$) in liquid exfoliation effectively produces TMD monolayers with binding multivalency for pathogenic bacteria. Density functional theory simulation reveals that the multivalent hydrogen bonding between dextran and TMD monolayers is very strong and thermodynamically favored ($\Delta E_b = -0.52$ eV). The resulting dextran/TMD hybrids (dex-TMDs) exhibit a stronger affinity ($K_d = 11$ nM) to *Escherichia coli* O157:H7 (*E. coli*) than *E. coli*-specific antibodies and aptamers. The dex-TMDs can effectively detect a single copy of *E. coli* based on their Raman signal.

[1] Department of Chemical Engineering, Hanyang University, Ansan 426-791, Republic of Korea. [2] Department of Chemical and Molecular Engineering, Hanyang University, Ansan 426-791, Republic of Korea. These authors contributed equally: Tae Woog Kang, Juhee Han, Sin Lee, In-Jun Hwang. Correspondence and requests for materials should be addressed to S.U.L. (email: sulee@hanyang.ac.kr) or to J.-H.K. (email: kjh75@hanyang.ac.kr)

Two-dimensional (2D) transition metal dichalcogenides (TMDs)[1–3] have emerged as a new class of nanomaterials owing to their unique physicochemical properties and potential applications in a variety of research fields including catalysis[4–6], optoelectronics[7], biosensing[8], and energy storage and conversion[9]. In order to fully utilize the outstanding properties of TMDs in the application of interest, monolayered or thin-layered TMD nanosheets are desirable as they exhibit layer-number-dependent electronic structures[10,11]. Hence, numerous efforts have been made to develop methods for obtaining thin-layered TMD nanosheets such as mechanical exfoliation[12,13], chemical vapor deposition[14,15], and liquid exfoliation[16]. Among them, liquid exfoliation has many advantages in terms of high production yield, mild condition, and low cost[16,17]. Most of the effective liquid exfoliation methods, however, require organic solvents with surface energies close to those of TMD nanosheets.

For biological, medical, and sensing applications and aqueous-phase catalysis, however, effective aqua exfoliation of TMD nanosheets without deformation of their electronic structures[18] is essential. To date, several approaches using surfactants[19,20], polymers[21–23], DNA[24], and a protein[25] have been developed, but it is still challenging to effectively exfoliate bulk TMDs into TMD monolayers in aqueous solutions and maintain them in a stably dispersed state without aggregation during reactions or assays. In particular, it remains more challenging to impart a specific function to TMD monolayers at the same time as their exfoliation and dispersion during an exfoliation process. The simultaneous exfoliation and functionalization strategy would help reduce efforts required for further modification of TMD nanosheets and improve their performance in the applications of interest. In order to control the simultaneous exfoliation and functionalization of TMD nanosheets in aqueous solutions, it is necessary to investigate and understand the thermodynamic properties of the interactions between TMD nanosheets and exfoliating molecules. The insight into the interactions at the TMD interfaces would enable us to effectively design TMD nanosheets with a specific function, which could be extended to the preparation of a variety of functional 2D nanomaterials. However, the precise understanding of the interactions between the TMD nanosheets and exfoliating molecules in aqueous solutions has been overlooked in most of the previous studies.

Pathogenic microorganisms are considered one of the most dangerous elements causing food-borne, water-borne, and air-borne diseases[26,27]. Many bacterial pathogens exhibit very low infectious concentrations, which is as low as ten cells for *Escherichia coli* O157: H7 (*E. coli* O157:H7) and *Salmonella*[28]. One of the most important events occurring during bacterial adhesion and infection is multivalent carbohydrate–lectin interaction[29,30]. Hence, it is essential to control these multivalent interactions for devising the strategies for bacteria detection and prevention of infection. Furthermore, the multivalent carbohydrate interactions can be employed for the simultaneous exfoliation and functionalization of TMD nanosheets in aqueous solutions and for bacteria detection.

The most widely applied method for bacteria detection is cultivation-based counting, which is, however, very time-consuming and exhibits low sensitivity[31,32]. For more sensitive and faster detection of bacteria, polymerase chain reaction (PCR)[33,34] and antibody-based immunoassays[35,36] have been suggested. PCR-based methods also require multiple sample preparation steps and professional skills, although they are sensitive in bacteria detection[37]. Antibody-based immunoassays exhibit better accessibility to point-of-detection for bacteria as compared to PCR; however, antibodies are expensive and susceptible to perturbation in environmental conditions and they have variable affinities to target antigens, which leads to a diminution in the reproducibility and sensitivity of the assay[28]. Therefore, there is still an increasing demand for a sensing method that achieves the sensitive, selective, rapid, and simple detection of pathogenic bacteria without the use of antibodies[38].

Herein, we report an effective strategy for the simultaneous exfoliation and functionalization of TMD nanosheets in an aqueous solution via multivalent hydrogen bonding of a carbohydrate polymer dextran for the sensitive optical detection of bacteria without the use of antibodies. The thermodynamic binding energy of the multivalent interaction between TMD nanosheets and dextran is also investigated using density functional theory (DFT) calculation. The as-prepared TMD nanosheets with glucan multivalency exhibit a specific recognition capability for *E. coli*, and are subsequently employed as an optical biosensor for the simple, rapid, and sensitive detection of bacteria in a single copy.

## Results

**Preparation of TMD nanosheets with glucan multivalency.** Thin TMD nanosheets ($WS_2$, $WSe_2$, and $MoSe_2$) were exfoliated from bulk TMDs and simultaneously functionalized using a carbohydrate polymer dextran MW 40,000 (denoted "dex") in an aqueous solution via a simple pulsed sonication process (Fig. 1a), resulting in dextran/TMD hybrids (denoted as "dex-TMDs"). As investigated in this work, the multivalent hydrogen bonding between the hydroxyl groups of dextran and the chalcogens (S or Se) of TMDs plays an important role in the aqueous phase exfoliation and functionalization of TMD nanosheets. Figure 1b shows that none of $WS_2$, $WSe_2$, and $MoSe_2$ nanosheets was exfoliated in the absence of dextran in the aqueous solution. However, all the three types of TMD nanosheets were effectively exfoliated in the presence of dextran in the aqueous solution (0.32 mg mL$^{-1}$) as shown in Fig. 1c. In order to demonstrate the importance of multivalent hydrogen bonding in the aqueous phase exfoliation of TMDs, glucose, which is a repeating unit in dextran, was added to a bulk TMD aqueous solution and sonicated under the same condition as dextran. As shown in Fig. 1d, no TMD nanosheets were obtained in the presence of glucose. As the hydrogen bonding multivalency of glucose is two hundred times smaller than that of dextran, the interaction of TMDs with glucose might be too weak to overcome the van der Waals interaction between the TMD interlayers for re-stacking. Furthermore, polyethylene glycol dimethyl ether (PEG, MW 35,000), which cannot undergo hydrogen bonding with TMDs, was added during the exfoliation process (see the Supplementary Methods). As expected, TMD nanosheets were hardly exfoliated in the presence of PEG in the aqueous solution (Fig. 1e). These results clearly indicate that the multivalent hydrogen bonding of dextran enables the effective exfoliation and functionalization of TMD nanosheets in an aqueous solution.

Subsequently, the effect of dextran concentration on the efficiency of exfoliation was investigated. After sonication and centrifugation of the $WS_2$ solutions containing different amounts of dextran, the absorbance of the obtained solutions at 297 nm was measured to quantify the amount of exfoliated $WS_2$ nanosheets[22]. When the weight ratio of dextran decreased from 1:3 (bulk TMD:dextran) to 15:1, the exfoliation efficiency of dex-$WS_2$ nanosheets increased (Supplementary Fig. 1). However, as the weight ratio of dextran became less than 15:1, the exfoliation efficiency of dex-$WS_2$ decreased. The amount of dex-$WS_2$ nanosheets was additionally quantified by inductively coupled plasma-atomic emission spectroscopy (ICP-AES) (Supplementary Fig. 1c). The tendency of the exfoliation efficiency measured by ICP-AES over the weight ratios was in good agreement with the UV absorbance-based plot. The highest concentration of dex-$WS_2$ nanosheets was obtained at a ratio of 15:1.

The as-prepared dex-TMD nanosheets were subsequently analyzed using transmission electron microscopy (TEM) and atomic force microscopy (AFM). The TEM images of dex-$WS_2$,

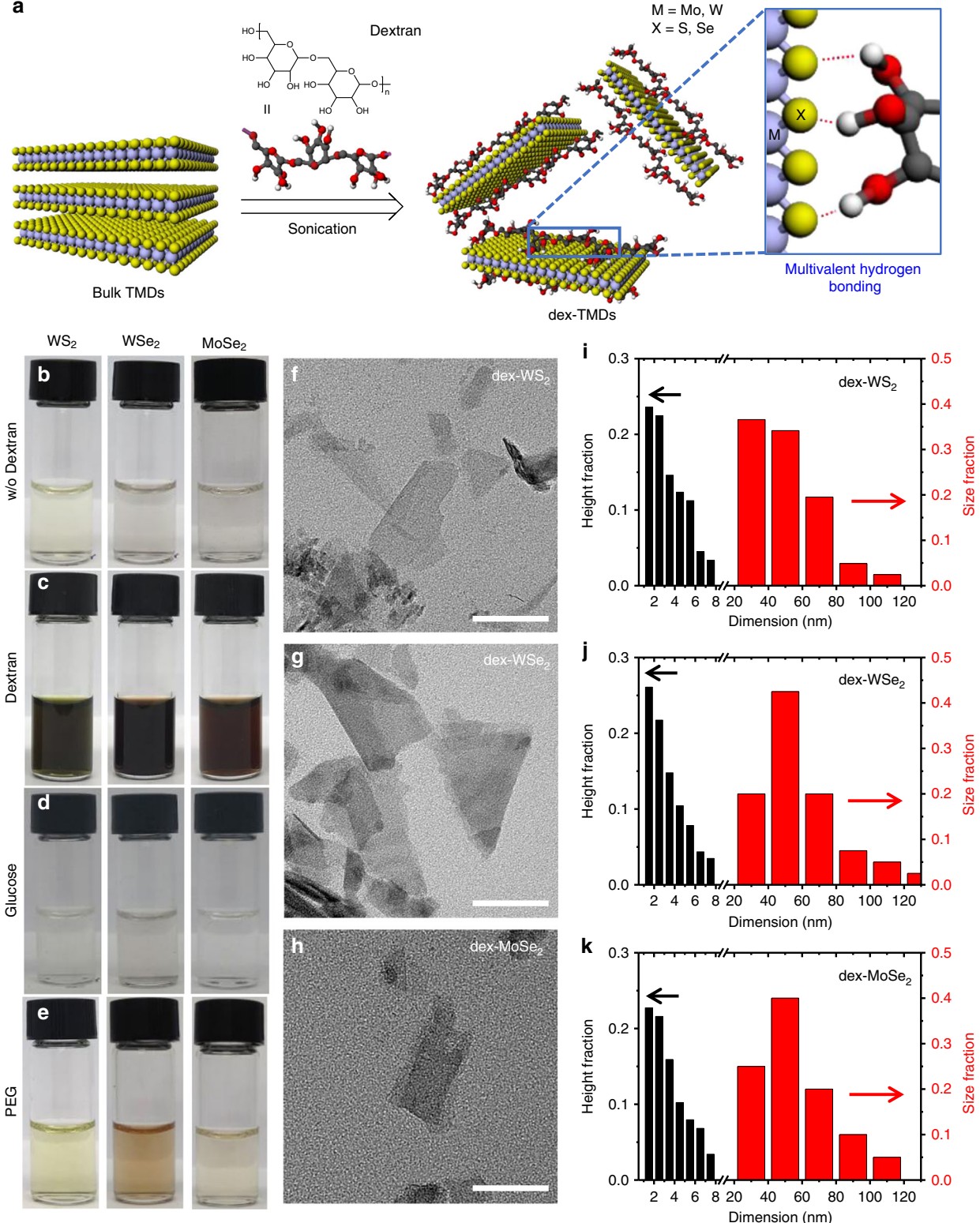

**Fig. 1** Simultaneous exfoliation and functionalization of TMDs with dextran. **a** Schematic illustration of the exfoliation and functionalization of TMDs via multivalent hydrogen bonding in an aqueous solution. Photographs of the solution of TMDs exfoliated **b** without dextran, **c** with dextran, **d** with glucose, and **e** with PEG. TEM images of **f** dex-WS$_2$, **g** dex-WSe$_2$, and **h** dex-MoSe$_2$ (scale bar, 50 nm). Height and lateral size profiles of **i** dex-WS$_2$, **j** dex-WSe$_2$, and **k** dex-MoSe$_2$ obtained using AFM

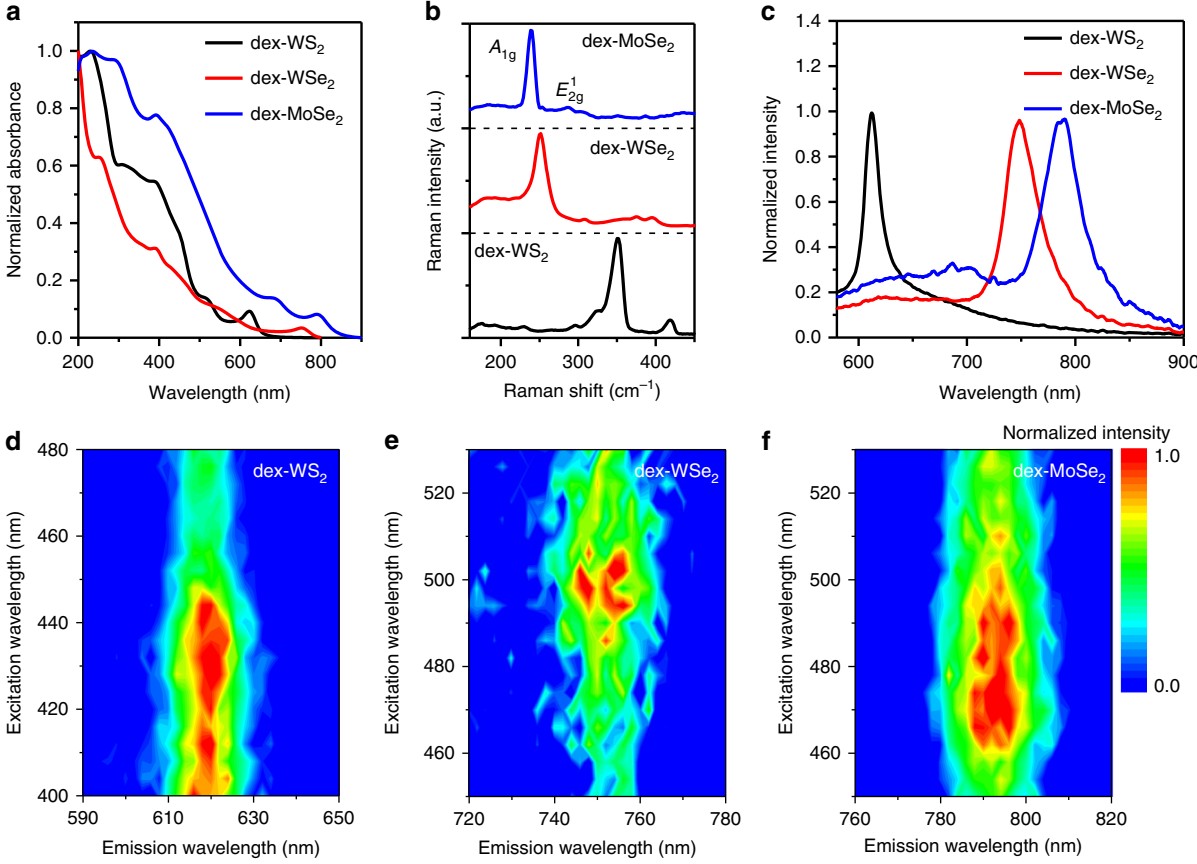

**Fig. 2** Optical properties of dex-TMDs. **a** UV–Vis extinction spectra, **b** Raman spectra, and **c** photoluminescence spectra of dex-WS$_2$, dex-WSe$_2$, and dex-MoSe$_2$ nanosheets. A 532-nm excitation laser was used. Excitation/emission profiles of **d** dex-WS$_2$, **e** dex-WSe$_2$, and **f** dex-MoSe$_2$, demonstrating excitation wavelength-independent photoluminescence

dex-WSe$_2$, and dex-MoSe$_2$ (Fig. 1f–h) show that they have a thin layer of 2D nanostructures with an average lateral size of 52 nm (Fig. 1i–k and Supplementary Fig. 2). The lattice structures and selected-area electron diffraction (SAED) patterns of dex-TMDs clearly show that they retain an intrinsic 2H phase after exfoliation and functionalization by dextran (Supplementary Fig. 3). After removing unbound dextran from the bulk solution via centrifugation, the morphology and thickness of dex-TMDs were measured using AFM. The morphology of dex-WS$_2$, dex-WSe$_2$, and dex-MoSe$_2$ appeared rough owing to dextran adsorption on the basal plane of TMDs via multivalent hydrogen bonding (Supplementary Fig. 4). Furthermore, the height histograms of dex-TMDs show that the thickness of dex-TMDs was higher than that of monolayer TMDs (Fig. 1i–k), revealing that 2D TMD nanosheets with multivalent glucan were successfully obtained.

**Optical properties of dex-TMD nanosheets**. The optical properties of dex-TMD nanosheets exfoliated in an aqueous solution were investigated. Figure 2a shows that the characteristic A excitonic absorption was clearly observed at 621, 752, and 790 nm for dex-WS$_2$, dex-WSe$_2$, and dex-MoSe$_2$, respectively, indicating that all of them maintain semiconducting electronic structures after exfoliation and functionalization[39]. The solution of TMDs exfoliated in the absence of dextran did not produce any characteristic A excitonic absorption peaks (Supplementary Fig. 5), indicating that no TMD nanosheets were obtained. To confirm edge and confinement effects on the absorption of dex-WS$_2$[20,40], dex-WS$_2$ was obtained at various centrifugation rates, and then its extinction spectra was taken (Supplementary Fig. 6). After normalization of the extinction spectra with the concentration of dex-

WS$_2$ measured by ICP-AES, the extinction coefficient at 235 and 290 nm was plotted as a function of centrifugation rates. The extinction coefficient at 235 nm was independent of the centrifugation rate, whereas the extinction coefficient at 290 nm decreased with increasing the centrifugation rate. It was also found that the wavelength of the A excitonic absorption of dex-WS$_2$ blue-shifted as the centrifugation rate increased. This result reveals that the edge and confinement effects are valid for dex-WS$_2$.

Figure 2b shows the characteristic Raman spectra of dex-WS$_2$, dex-WSe$_2$, and dex-MoSe$_2$. In the Raman spectra of dex-WS$_2$ (black line), two distinct peaks for a combination of in-plane vibrational mode ($E^1_{2g}$) and longitudinal acoustic mode (2LA(M)), and an out-of-plane vibrational mode ($A_{1g}$) appeared at 350 and 418 cm$^{-1}$, respectively[41]. As the $A_{1g}$ mode of WS$_2$ is more significantly dependent on its layer number than the 2LA(M) mode, their intensity ratio can be used as an indicator to determine the number of WS$_2$ layers[42]. The intensity ratio of the $A_{1g}$ peak to the 2LA(M) peak in the Raman spectrum of dex-WS$_2$ was 6.7, indicating that monolayer dex-WS$_2$ was mainly exfoliated by dextran. Exfoliated dex-WSe$_2$, however, showed only a single Raman signal at 251.7 cm$^{-1}$ (red line), which is consistent with the result of WSe$_2$ monolayer reported in other literature[43]. This singlet is considered to be generated by the degeneration of $A_{1g}$ and $E^1_{2g}$ modes as the number of WSe$_2$ decreases to monolayer or a few layers. In the case of MoSe$_2$, it is known that the peak position of the $A_{1g}$ mode depends on the number of MoSe$_2$ layers, that is, 240 cm$^{-1}$ for monolayer and the wavenumber increases as the number of layers increases[43]. The $A_{1g}$ mode of exfoliated dex-MoSe$_2$ strongly appeared at 239.5 cm$^{-1}$ (blue line, Fig. 2b), suggesting that a majority of dex-MoSe$_2$ nanosheets were monolayers.

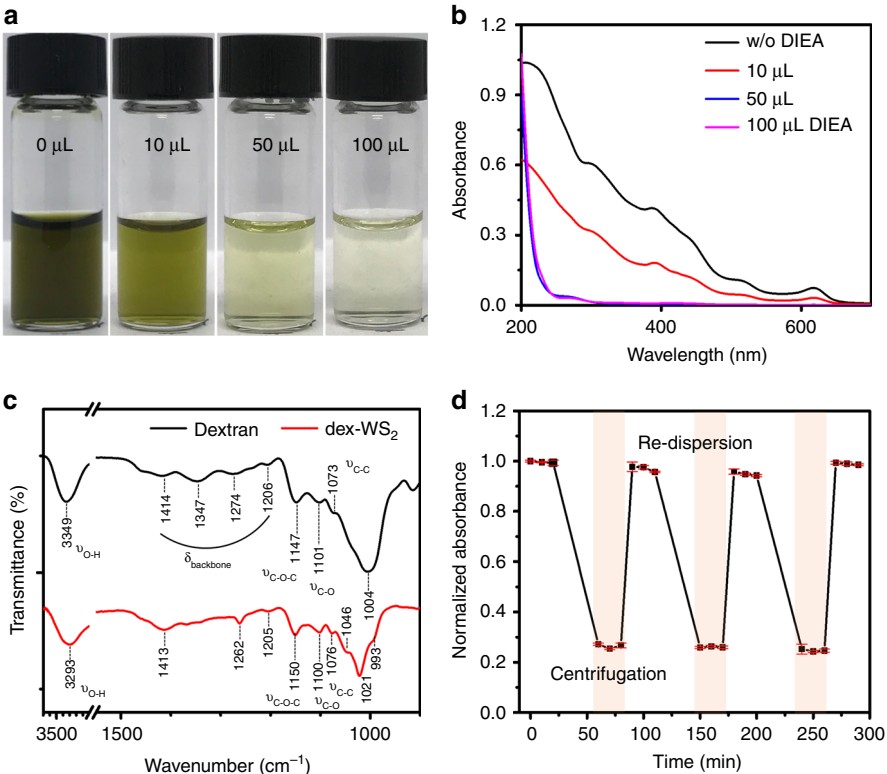

**Fig. 3** Multivalent hydrogen bonding in dex-WS$_2$ nanosheets. **a** Photographs and **b** UV–Vis spectra of the solution of WS$_2$ nanosheets exfoliated and functionalized by dextran in the presence of a proton acceptor DIEA at various concentrations. **c** FT-IR spectra of dex-WS$_2$ and dextran, showing hydrogen bonding between WS$_2$ and dextran. **d** Absorbance plot of the repeated redispersion of dex-WS$_2$ after centrifugation, demonstrating that the multivalent hydrogen bonding is sufficiently strong to maintain the stability of the dex-WS$_2$ colloid in repeated centrifugation-redispersion processes. The absorbance was measured at 621 nm. All error bars represent a standard deviation from the mean values ($n = 4$)

Subsequently, the photoluminescence (PL) emission of dex-WS$_2$, dex-WSe$_2$, and dex-MoSe$_2$ was investigated as shown in Fig. 2c. As compared to bulk WS$_2$, strong PL emission of dex-WS$_2$ appeared at 612 nm (black line, Fig. 2c), showing the transformation of an indirect semiconductor of bulk WS$_2$ into a direct semiconductor of WS$_2$ monolayer. The exfoliated dex-WSe$_2$ emitted a single PL peak at 748 nm, which distinctly suggests that a majority of WSe$_2$ nanosheets exfoliated by dextran via multivalent hydrogen bonding were monolayers with a direct band gap (red line). According to previously reported results, two and three layers of WSe$_2$ exhibited two PL emission peaks at wavelengths longer than 748 nm[20,43]. In addition to dex-WS$_2$ and dex-WSe$_2$, dex-MoSe$_2$ also exhibited characteristic PL emission at 790 nm for monolayers (blue line). In the previous studies, two and three layers of MoSe$_2$ exhibited their PL emission at wavelengths longer than 800 nm. Subsequently, we measured the excitation and emission profiles of dex-WS$_2$, dex-WSe$_2$, and dex-MoSe$_2$ (Fig. 2d–f). All the dex-TMDs emitted excitation wavelength-independent PL emission with high intensities. Furthermore, it was observed that each dex-TMD had its own excitation wavelength at which its PL emission was the most intense.

As reported in the previous literature[20], we tried to estimate the monolayer volume fraction of dex-WS$_2$ by applying its PL/Raman intensity ratio to the reported metric. However, the estimated monolayer volume fraction of dex-WS$_2$ was much lower than what was expected based on its AFM height profile and PL/Raman properties. As discussed in the PL and Raman spectra of dex-WS$_2$ above, a majority of dex-WS$_2$ should be monolayers. There are two expected reasons why the previous metric based on a PL/Raman intensity ratio for monolayer volume fractions was not valid for dex-WS$_2$. First, the PL intensity of WS$_2$ nanosheets can be affected by an exfoliating molecule. The quantum yield

(QY) of dex-WS$_2$ might be different from that of surfactant-exfoliated WS$_2$ nanosheets. Second, the QY of WS$_2$ nanosheets can be influenced by the extent of defects in which non-radiative recombination takes place, resulting in a decrease in the QY.

All the aforementioned optical properties of dex-TMDs demonstrate that bulk TMDs with indirect band gaps were effectively exfoliated into TMD monolayers with direct band gaps via multivalent hydrogen bonding between the hydroxyl groups of dextran and the chalcogens of TMDs in the aqueous solution.

**Verification of multivalent hydrogen bonding in dex-TMDs.** We conducted further experiments to verify that the multivalent hydrogen bonding of dextran is a primary factor in the effective exfoliation and functionalization of TMD nanosheets in an aqueous solution. First, a strong hydrogen acceptor, *N,N*-diiso-propylethylamine (DIEA), which can prevent dextran from undergoing multivalent hydrogen bonding with TMDs, was added to the bulk WS$_2$ solution, followed by sonication. As the concentration of DIEA increased, the color of the exfoliated WS$_2$ solution collected after centrifugation became lighter and more transparent (Fig. 3a), indicating that the amount of exfoliated WS$_2$ nanosheets significantly decreased. Furthermore, the char-acteristic A1 and B1 excitonic transitions for WS$_2$ monolayers at 621 and 512 nm significantly decreased in the extinction spectra of the exfoliated solutions obtained in the presence of DIEA, and eventually disappeared as 50 μL of DIEA was added (Fig. 3b and Supplementary Fig. 7). This experimental result apparently shows that the multivalent hydrogen bonding of dextran plays a very important role in the exfoliation of TMDs in an aqueous solution.

Subsequently, we investigated the effect of the chain length of dextran on the exfoliation efficiency of WS$_2$. When dextran

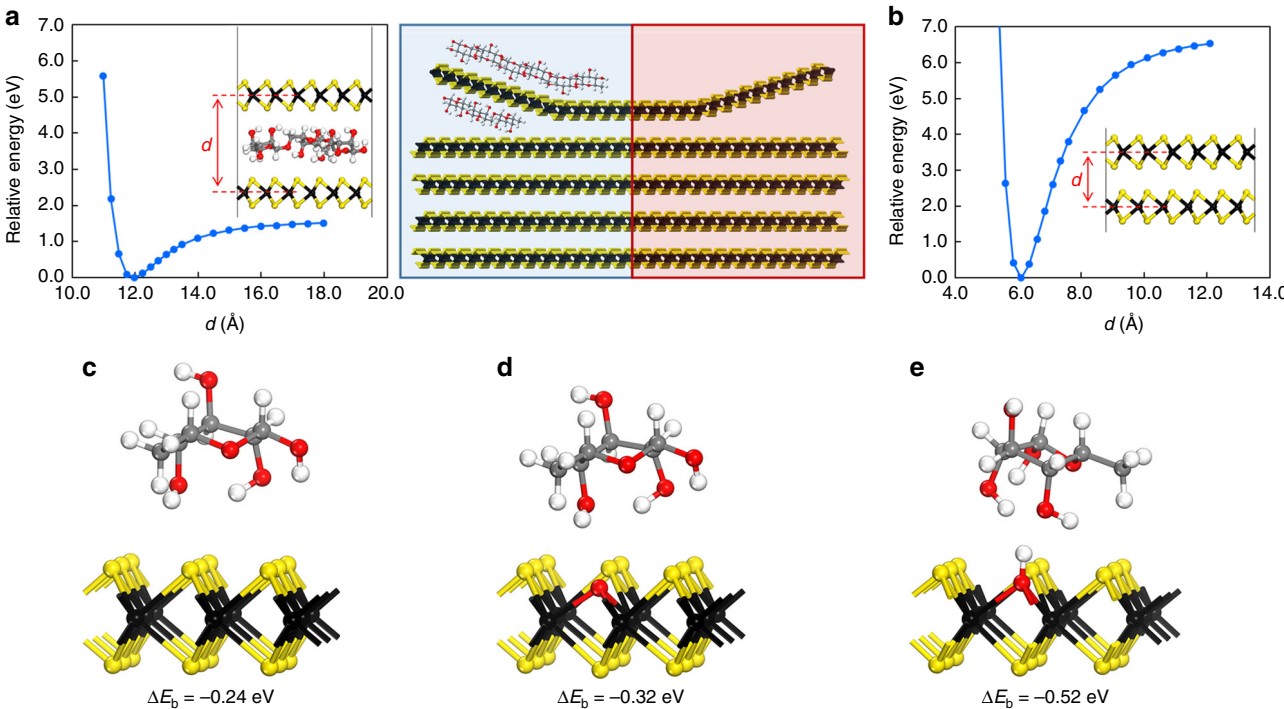

**Fig. 4** DFT simulation for the interactions between dextran and TMD. Potential energy of interaction between two parallel monolayers of $WS_2$ **a** with and **b** without dextran. The central image illustrates an exfoliation process in two cases. Binding energy of the hydrophilic hydroxyl groups of dextran with **c** pure $WS_2$, **d** oxygen-defective $WS_2$, and **e** hydroxyl-defective $WS_2$

of MW 1000 (dex1000) was employed in an exfoliation process (Supplementary Fig. 8a), the color of the obtained solution of exfoliated $WS_2$ (dex1000-$WS_2$) became lighter and more transparent than the solution of $WS_2$ nanosheets exfoliated by dextran of MW 40,000 (dex-$WS_2$), indicating that the amount of exfoliated $WS_2$ nanosheets was reduced as the molecular weight of dextran decreased. Furthermore, two excitonic transitions in the extinction spectrum of dex1000-$WS_2$ exhibited lower intensity than in the spectrum of dex-$WS_2$ (Supplementary Fig. 8b). As discussed, when the chain length was reduced to the extent of a monomer unit (glucose), no TMD nanosheets were exfoliated (Fig. 1d). These results reveal that the exfoliation efficiency of $WS_2$ nanosheets decreased as the hydrogen bonding multivalency of dextran was reduced by shortening its chain length.

The multivalent hydrogen bonding of dextran with $WS_2$ nanosheets was directly observed using Fourier transform infrared (FT-IR) spectroscopy. As shown in Fig. 3c, the O–H stretching mode of the hydroxyl groups of dextran in dex-$WS_2$ was shifted toward a lower frequency (3293 cm$^{-1}$) as compared to that (3349 cm$^{-1}$) of pristine dextran. This peak shift is attributed to the fact that the OH bond strength becomes weak owing to hydrogen bonding with $WS_2$. Furthermore, the C–O stretching modes of pristine dextran appeared very broad at approximately 1004 cm$^{-1}$ owing to intramolecular hydrogen bonding[44]. However, the C–O stretching modes in dex-$WS_2$ nanosheets appeared sharp and were more clearly resolved at 1046, 1021, and 993 cm$^{-1}$, because the dextran on the basal plane of $WS_2$ mostly underwent intermolecular hydrogen bonding—rather than intramolecular hydrogen bonding—with the sulfur atoms. In the X-ray photoelectron spectroscopy spectra of dex-$WS_2$ (Supplementary Fig. 9), the characteristic peaks of dextran were also observed, demonstrating that dex-$WS_2$ hybrids were successfully prepared. Moreover, the W4f peaks of dex-$WS_2$ confirmed that it retained an intrinsic 2H phase after exfoliation and functionalization. It was observed that dex-$WS_2$ was partially oxidized to produce tungsten oxide during a process of exfoliation. The tungsten oxide content

in dex-$WS_2$ increased to 31% as compared to the proportion (15%) of starting bulk $WS_2$. However, the tungsten oxide content in dex-$WS_2$ significantly decreased to 21% as the sonication amplitude was reduced to a half.

Subsequently, the stability of multivalent hydrogen bonding in dex-$WS_2$ nanosheets was investigated by measuring their colloidal stability. After removing unbound dextran from the bulk solution, dex-$WS_2$ underwent a repeated centrifugation and re-dispersion process. The A excitonic absorption at 621 nm was then measured at each stage to determine the colloidal stability of dex-$WS_2$ (Fig. 3d). The absorption of dex-$WS_2$ remained constant even after repeated centrifugation and re-dispersion, indicating that dex-$WS_2$ maintained the stability of its hybrid structure without aggregation. Furthermore, dex-$WS_2$ was stably dispersed without re-aggregation for 2 months. These results suggest that multivalent hydrogen bonding in dex-$WS_2$ is sufficiently strong to maintain the stability of the hybrid nanostructure in an aqueous solution for a certain duration.

**DFT calculation for the interaction between dextran and TMDs.** We conducted computational DFT simulation to calculate the thermodynamic properties of the interactions between dextran and TMD nanosheets. First, we calculated the interaction potential energy between two parallel monolayers of $WS_2$ in the presence or absence of dextran as a function of the interlayer distance. As shown in Fig. 4a, b, the interaction potential energy of the $WS_2$ interlayer was noticeably lower in the presence of dextran during the exfoliation process in the aqueous solution as compared to the value calculated in the absence of dextran. This reduced potential energy clearly reveals that the exfoliation of TMD nanosheets in the aqueous solution can be facilitated by the incorporation of dextran and they can be stably dispersed with lower tendency toward re-aggregation. Accordingly, the required energy for the exfoliation of $WS_2$ monolayer in the presence of dextran is significantly reduced from 5 to 1 eV for an increase in the interlayer distance of 2 Å from the most stable distance.

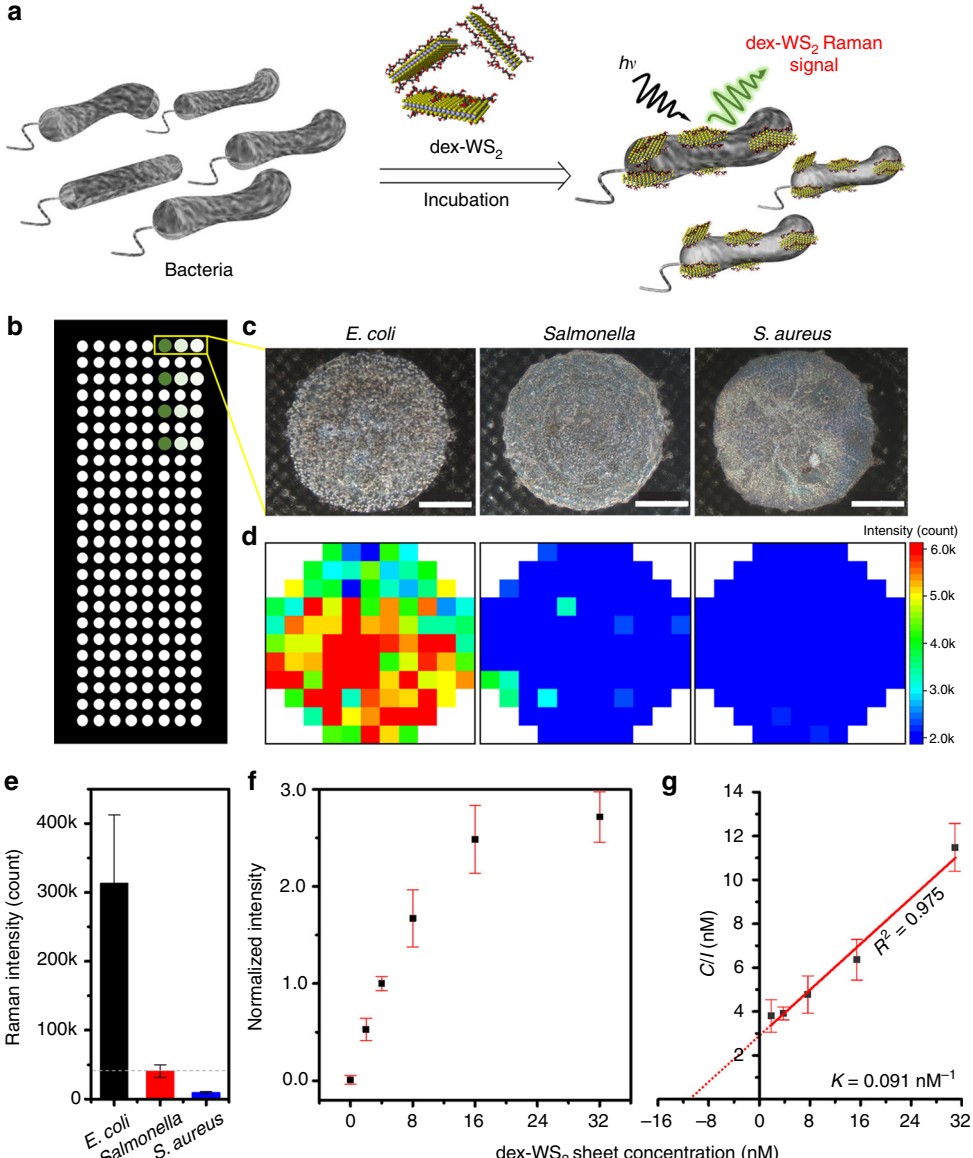

**Fig. 5** Antibody-free optical detection of bacteria with dex-WS$_2$. **a** Schematic illustration of the antibody-free optical detection of bacteria with dex-WS$_2$. **b** Glass microarray and **c** Optical photographs of *E. coli*, *Salmonella typhimurium*, and *S. aureus* ($2\times10^6$ CFU) treated with dex-WS$_2$ (25 µg mL$^{-1}$) (scale bar, 500 µm). **d** Raman images of the dex-WS$_2$-treated bacteria based on the 2LA(M) peak of dex-WS$_2$ at 352 cm$^{-1}$, showing the selective recognition of *E. coli*. **e** The total intensity of the Raman signal of dex-WS$_2$ bound on three different types of bacteria. **f** *E. coli* detection as a function of dex-WS$_2$ concentrations for the extraction of its affinity to *E. coli*. **g** Langmuir isotherm for the binding of dex-WS$_2$ to *E. coli*. All error bars represent a standard deviation from the mean values ($n = 4$)

Subsequently, we calculated the binding energy of the hydrophilic hydroxyl groups of dextran on the surface of pure WS$_2$ or defective WS$_2$ monolayers. As shown in Fig. 4c, the binding energy of the hydrophilic part of dextran with pure WS$_2$ not bearing any defects was determined to be −0.24 eV, which is already thermodynamically favored. If a WS$_2$ nanosheet has an oxygen defect (Fig. 4d), the binding energy becomes a greater negative value (−0.32 eV), indicating that the interaction between dextran and the WS$_2$ nanosheet can be stronger. Furthermore, the binding energy of dextran with a hydroxyl-defective WS$_2$ nanosheet was determined to be a greater negative value (−0.52 eV) than that with an oxygen-defective WS$_2$ nanosheet (Fig. 4e). As these defective WS$_2$ nanosheets can be produced during the exfoliation process (Supplementary Fig. 9), three types of interactions between dextran and WS$_2$ nanosheets are likely to occur in the aqueous solution.

The simulation results clearly show that the hydroxyl groups of dextran can forge strong bonds with pure and defective WS$_2$ monolayers, which is sufficient to exfoliate and stably disperse them without re-aggregation in the aqueous solution. In particular, the exfoliation of TMD monolayers can be further facilitated by the multivalent binding of dextran, which leads to much stronger interactions in the TMD hybrids as observed in carbohydrate–lectin interactions.

**Antibody-free recognition of pathogenic bacteria.** *E. coli* O157: H7 is a Gram-negative pathogen with carbohydrate binding sites such as Type 1 fimbriae (*fim*) and pyelonephritis-associated pili (*pap*) containing the *fim*H protein[45,46]. The *fim*H lectin is known as a mannose-binding protein ($K_d = 2.3$ µM), but it also has a very low binding affinity to glucose ($K_d = 9.4$ mM). The low binding affinity of this lectin to glucose inspires us to design

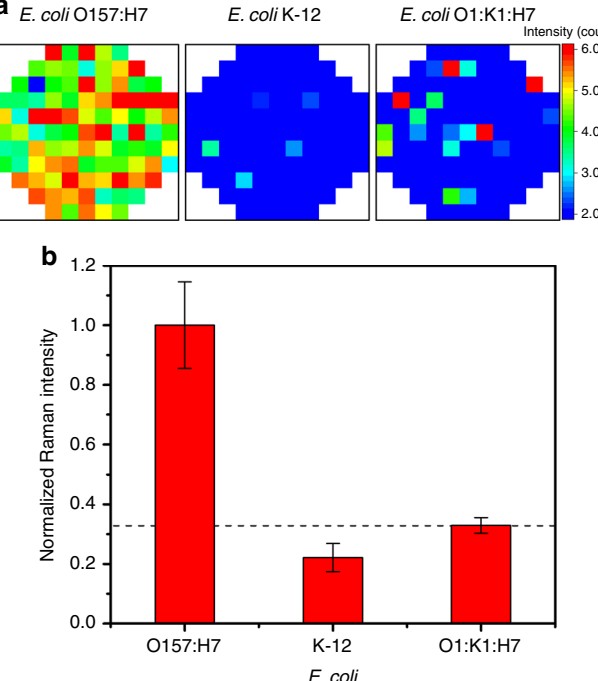

**Fig. 6** Recognition selectivity of dex-WS₂ to different strains of *E. coli*. **a** Raman images of the dex-WS₂-treated bacteria based on the 2LA(M) peak of dex-WS₂ at 352 cm⁻¹, showing the effective recognition of *E. coli* O157:H7 against *E. coli* K-12 and *E. coli* O1:K1:H7. **b** The normalized total intensity of the 2LA(M) Raman signal of dex-WS₂ bound on three different strains of *E. coli*. All error bars represent a standard deviation from the mean values ($n = 4$)

dex-TMD nanosheets with glucan multivalency for improved and antibody-free recognition of *E. coli*. We hypothesize that dex-TMDs with glucose multivalency would enable selective recognition of *E. coli* O157:H7 with a significantly enhanced binding affinity against other pathogenic bacteria.

In order to confirm our hypothesis, three different types of pathogenic bacteria—*E. coli* O157:H7, *Salmonella typhimurium* (Gram-negative), and *Staphylococcus* (*S.*) *aureus* (Gram-positive)—were incubated with dex-WS₂ in a separate reaction vessel for 1 h (Fig. 5a, see the Supplementary Methods for bacteria culture). After washing each bacteria with phosphate-buffered saline (PBS) to remove unbound dex-WS₂, the optical photograph of a pile of bacteria centrifuged was taken (Supplementary Fig. 10a). The dex-WS₂-treated *E. coli* turned light yellow from white after reaction with dex-WS₂, indicating that dex-WS₂ bound on the surface of the bacteria. However, *Salmonella typhimurium* and *S. aureus* retained an original white color even after reaction with dex-WS₂, indicating that dex-WS₂ hardly bound on these bacteria. Then, the dex-WS₂-treated bacteria ($2 \times 10^6$ colony-forming unit (CFU)) were placed on glass microarrays for detection by measuring the Raman scattering signals (2LA (M), $E^1_{2g}$ and $A_{1g}$) of dex-WS₂ bound on the bacteria (Fig. 5b, c). As shown in the results of ensemble measurement (Fig. 5d), strong Raman signals of dex-WS₂ were observed only from *E. coli*. Negligible Raman signals were measured from both *Salmonella typhimurium* and *S. aureus* as shown in the Raman images generated by the intensity of the 2LA(M) peak of dex-WS₂. The total intensity of dex-WS₂ obtained from the three types of bacteria demonstrates that dex-WS₂ with glucan multivalency could selectively recognize *E. coli* against *Salmonella typhimurium* and *S. aureus* (Fig. 5e). The content of dex-WS₂ bound on the bacteria was further quantified using ICP-AES

(Supplementary Fig. 10b). A large amount of WS₂ (6.95 μg mL⁻¹) was measured from the dex-WS₂-treated *E. coli*, whereas a trace amount of WS₂ was observed from the dex-WS₂-treated *Salmonella typhimurium* and *S. aureus*. This is in good agreement with the result of the bacterial detection based on the Raman signal measurement on glass microarrays. This result reveals again that dex-WS₂ was able to recognize *E. coli* without the use of antibodies.

Notably, the Raman signals (2LA(M) and $A_{1g}$) of dex-WS₂ remained very intense with almost the same intensity even after binding on *E. coli* whereas its fluorescence was completely quenched (Supplementary Fig. 11). This fluorescence quenching of dex-WS₂ might be caused by the energy transfer to a variety of proteins on *E. coli* or the charge transfer to K⁺ ions released out of the bacteria as previously observed in quasi-2D MoS₂ adsorbed on yeast cells[47].

To investigate further the selectivity of dex-WS₂ to *E. coli* O157:H7, other strains of *E. coli* such as *E. coli* K-12 and *E. coli* O1:K1:H7 were incubated with dex-WS₂, and then the 2LA(M) Raman signals at 352 cm⁻¹ were collected from the bacteria on glass microarrays. As shown in Fig. 6, the strong Raman signals of dex-WS₂ were observed only from *E. coli* O157:H7. The very weak Raman signals of dex-WS₂ were measured from both *E. coli* K-12 and *E. coli* O1:K1:H7. This result clearly reveals that dex-WS₂ was able to effectively recognize *E. coli* O157:H7 against other strains of *E. coli*. It is worth noting that dex-WS₂ could differentiate *E. coli* O157:H7 from other strains of *E. coli* K-12 and *E. coli* O1:K1:H7, although they all have *fim*H on their membrane[48,49]. We speculate that the increased binding affinity of dex-WS₂ to *E. coli* O157:H7 might be ascribed to the fact for the slight structural difference of the *fim*H on the *E. coli* or the recognition capability of a three-dimensional structure of dextran created on the rigid surface of WS₂ nanosheets.

We calculated the binding affinity of dex-WS₂ to *E. coli*. As shown in Fig. 5f, the Raman signal of dex-WS₂ from *E. coli* gradually increased when its concentration increased in the reaction with the bacteria, indicating that the binding kinetics of dex-WS₂ exhibit a concentration dependency. This concentration-dependent response was subsequently fitted with the Langmuir isotherm to extract the dissociation constant ($K_d$) of dex-WS₂ for binding with *E. coli* (Fig. 5g, see the details in the Supplementary Methods). The obtained $K_d$ value was 11 nM, which is much lower than the reported values for a monoclonal antibody, an aptamer specific for the bacteria[50,51], and mannose. This lower $K_d$ value suggests that dex-WS₂ with glucan multivalency has a higher binding affinity to *E. coli* than the antibody, aptamer, and mannose, which leads to the sensitive recognition of the target bacteria. We additionally measured the binding affinity of dex-WS₂ and an *E. coli*-specific antibody using a quartz crystal microbalance (QCM), and compared their binding affinity to *E. coli* O157:H7 (Supplementary Fig. 12, see the details in the Supplementary Methods). The QCM measurement reveals again that dex-WS₂ exhibited a higher binding affinity to *E. coli* O157:H7 than the *E. coli*-specific antibody. This significantly enhanced binding affinity of dex-WS₂ can be attributed to its multivalent binding ability for lectins on *E. coli*.

Subsequently, a single copy of bacteria was detected with dex-WS₂ (see the Supplementary Methods). After the reaction of *E. coli*, *Salmonella typhimurium*, or *S. aureus* with dex-WS₂, very dilute bacteria solution was spotted on microarrays for the Raman imaging based on its 2LA(M) peak at 352 cm⁻¹. As shown in Fig. 7, a very intense Raman signal of dex-WS₂ was observed from the single copy of *E. coli* whereas there was no noticeable Raman signal from both *Salmonella typhimurium* and *S. aureus*. Furthermore, when the dex-WS₂-treated *E. coli* was analyzed using scanning electron microscopy, dex-WS₂ nanosheets were clearly observed on the surface of

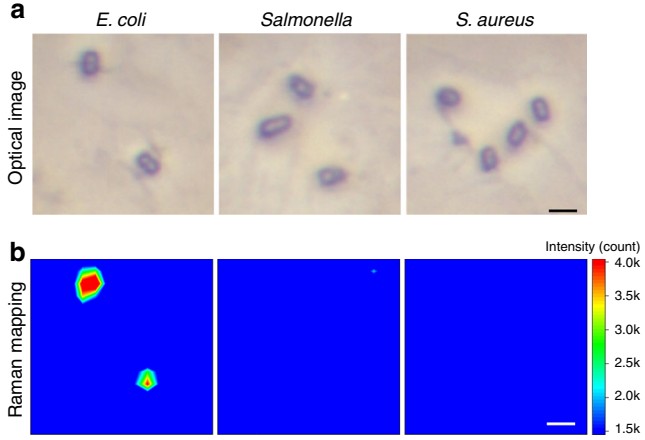

**Fig. 7** Detection of a single copy of bacteria with dex-WS₂. **a** Optical photographs of distinctly spread single bacterium. **b** Raman mapping of the single bacterium treated with dex-WS₂ based on the Raman scattering peak at 352 cm$^{-1}$, demonstrating that dex-WS₂ nanosheets could selectively detect *E. coli* at single copy level (scale bar, 2 μm)

the treated *E. coli*, whereas no nanosheets were observed on pristine *E. coli* (Supplementary Fig. 13). These results reveal that dex-WS₂ nanosheets with glucan multivalency were able to effectively detect *E. coli* pathogens in a single copy without the use of antibodies.

As compared to previous methods for bacteria detection, this sensing method based on dex-TMDs with glucan multivalency has many advantages: the sensing material dex-TMDs can be more readily prepared and are not significantly susceptible to perturbation in physical conditions. Furthermore, the dex-TMD sensing method enables the fast, simple, sensitive, and cost-effective detection of pathogenic bacteria.

## Discussion

Glucan multivalent interactions facilitated the simultaneous exfoliation and functionalization of TMD nanosheets in an aqueous solution, resulting in dex-TMD hybrids with a specific function for the selective recognition of *E. coli* without the use of antibodies. Experimental data and DFT simulation revealed that multivalent hydrogen bonding of dextran was a primary factor in the effective exfoliation and stable functionalization of TMD nanosheets in the aqueous solution. Furthermore, the multivalent hydrogen bonding of dextran with TMD nanosheets was observed to be very strong and thermodynamically favored. The as-prepared dex-TMDs were able to selectively detect a single copy of *E. coli* without the use of antibodies. In addition, the dex-TMDs could differentiate *E. coli* O157:H7 from other strains of *E. coli* such as *E. coli* K-12 and *E. coli* O1:K1:H7. This exfoliation and functionalization approach based on multivalent hydrogen bonding can be extended to other 2D nanomaterials to design diverse hybrids with new functions. Moreover, the antibody-free sensing method based on dex-TMDs with binding multivalency can evolve into nanosensors capable of both detection and therapy of various diseases.

## Methods

**Synthesis of TMD nanosheets with glucan multivalency**. In this study, dex-TMD nanosheets were obtained via liquid exfoliation of bulk TMDs in the presence of dextran (MW 40,000) in water. Accordingly, 6 g of bulk TMD (8.4 g for WSe₂) was added to 200 mL of water containing dextran (2 g L$^{-1}$), followed by sonication using a tip sonicator in an ice bath for 5 h at 75% amplitude with a pulse of 6 s on and 2 s off. The resulting solution was centrifuged at $1977 \times g$ for 1.5 h to discard unexfoliated TMDs. The supernatant was collected and centrifuged again at $15,344 \times g$ for 1.5 h. The sediment was collected, but the supernatant was discarded to remove unbound dextran and a smaller size of TMD particles. Subsequently, the sediment obtained at $15,344 \times g$ was re-dispersed in 25 mL of water and centrifuged

again at $3024 \times g$ for 1.5 h. The supernatant solution was finally collected to obtain dex-TMD monolayers ($3024$–$15,344 \times g$).

**DFT calculation for the interaction between dextran and TMDs**. All ab initio calculations were performed with the Vienna Ab initio Simulation Package (VASP 5.4.1)[52–55] using the projector augmented wave method[56,57] with the generalized gradient approximation based on the Perdew–Burke–Ernzerhof exchange-correlation functional[58,59]. In order to consider van der Waals interaction, the DFT + D3 method[59], in which the dispersion coefficients depend on the local structural geometry, was included in this work. Periodic boundary conditions were used in all directions and 20 Å of vacuum was used in the z-direction to separate the slabs. A plane-wave cutoff energy of 500 eV was employed to control the fineness of this mesh. All the calculations were carried out using fully optimized structures, where lattice vectors and ionic positions were fully relaxed until the maximum atomic forces were <0.04 eV Å$^{-1}$.

**General procedure for the detection of bacteria with dex-TMDs**. First, 100 μL of dex-TMD (125 μg mL$^{-1}$) was dispersed in 500 μL of PBS (10 mM, pH 7.4). Subsequently, 50 μL of bacteria was added to the dex-TMD solution, in which the final concentration of bacteria was $10^8$ CFU mL$^{-1}$. The resulting mixture was incubated for 1 h at 25 °C (500 rpm, Thermomixer). The reaction mixture was centrifuged for 5 min (4 °C, $1977 \times g$) to collect the bacteria and remove unbound dex-TMDs. The collected bacteria were thereafter washed with PBS several times to further remove unbound dex-TMDs in the solution. After the addition of 50 μL of PBS to the bacteria sediment, 2 μL of the bacteria solution ($2 \times 10^6$ CFU) was placed on glass microarrays for measurement.

**Data availability**. All the data supporting the findings of this work are available on the article and its Supplementary Information files and/or from the corresponding author upon request.

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

## Acknowledgements

This work was supported by the Samsung Research Funding Center of Samsung Electronics under project number SRFCTA1503-02.

## Author contributions

J.-H.K., T.W.K., J.H., S.L., and I.-J.H. conceived the main idea and developed this project. T.W.K., J.H., S.L., and I.-J.H. carried out most of the experiments and analyzed the data. S.-J.J., J.-M.J., and J.-K.Y. conducted the characterization of TMDs using Raman and fluorescence spectroscopy. M.-J.K. took the TEM images of TMDs. B.J., C.H.L., and S.U.L. carried out DFT simulation. J.-H.K., T.W.K., J.H., S.L., I.-J.H., and S.U.L. wrote the manuscript and Supplementary Information.
