## [Peer Review File · Nature Communications]

Reviewers' comments:

Reviewer #1 (Remarks to the Author):

This is a good work

The authors presented the exfoliation and surface functionalization of TMDs. They used dex to functionalise the surface and eventually produce selectivity to Salmonellae bacteria

The DFT calculations shows great affinity to Salmonellae complex that provides a selectivity

I support its publication

I only have a few minor comments:

1- Is there any reference for "WS2 mostly underwent 215 intermolecular hydrogen bonding—rather than intramolecular hydrogen bonding"

2- The first report on the incubation of TMD flakes on cells for their recognition is Nano Letters 14 (2), 857-863, 2014. Include and discuss it

3- What are the lateral dimensions of the flakes. Add it to figure 1. Is there any significance on the lateral dimension in terms of corners and edges effects in your work

4- I am not quite convinced that the reason for Raman signal selectivity for the last figure is what you presented. Are you able to add one more evidence (image, FTIR discussion, ...). The selectivity is remarkable though

Reviewer #2 (Remarks to the Author):

The authors reported a new and high efficient method to exfoliate two-dimensional transition metal dichalcogenides (TMDs) using dextran. The resulting TMD sheet were functionalized with dextran and could be used to selectively capture E. Coli and perform selective E. Coli detection. The authors presented a thorough study on the exfoliation conditions and characterization, together with a DFT calculations. It is a very sound paper, but in terms of the broad impact claimed by the authors, it is limited. I would suggest publishing the manuscript in a more specific journal than in Nature Communications. Below are my concerns:

1. According to the results demonstrated by the authors, the functionalization of the TMDs with dextran is a natural process of the exfoliation, it is not a designed functionalization process which claimed by the authors in Introduction, "a simultaneous exfoliation and functionalization strategy" (Page 3 Line 63). The application example is to use that nature of such a functionalized surface to capture E. Coli. Thus, the application of the materials is limited. Unless the authors could demonstrate other applications/or exfoliation/functionalization example based on this principle?

2. For the bacteria detection, it is also important to differentiate E. Coli O157:H7 from other E.coli as well. However, other E.Coli may also contains fimH protein on their membrane (see Nature Communication, 2016, 7, 10738).

3. Using dextran based polymer to exfoliate 2D materials has been reported before, see *Macromolecules*. 2015, Vol.48(18), p.6628-6637.
4. The authors using the UV-Vis absorbance to compare the amount of nanosheet extracted. Though in general this may indirectly reflect the amount of nanosheet extracted, but the optical property of nanosheet not only determined by the concentration, but also the size and shape of the nanosheet.
5. The authors claimed that the dex-TMDs have a better affinity to *E. Coli* O157:H7 than *E. Coli*-specified antibodies and aptamers. They referred to Refs. 45 & 46. It is questionable whether such a comparison is fair or not since the experiments conditions are very different. Such a claim can be concluded if the authors can design a convincing comparison experiment by considering the relative concentration of the capturing agents, the same bacteria, under the same operation conditions.

Reviewer #3 (Remarks to the Author):

The authors present a combined exfoliation-functionalisation approach to obtain colloiddally stable TMD dispersions in aqueous media. It is suggested that a multivalent hydrogen bonding between dextran and the TMDs is the source of this stability. The dispersions were characterised by a range of spectroscopic and microscopic techniques and showed good potential as biosensor in detection of bacteria. The study is interesting and overall thoroughly conducted. The binding motif- if it can be confirmed in future work – opens new possibilities to anchor diverse molecules to the surface of the nanosheets which will be of great impact. In general, the manuscript seems suitable for publication in *Nature Comm.* after the following points are addressed.

- The authors discuss that defects on the TMDs may further help the binding. While this seems valid from the theoretical viewpoint, it is nonetheless a bit worrying that the oxide content according to XPS is usually high. The authors should test whether these defects are introduced on exfoliation via sonication or already present in the starting material. If they are introduced by the sonication, the authors should minimize this impact for example by lowering the amplitude to avoid damage to the sheets. Otherwise, the question remains whether it would also work for defect-free nanosheets.
- The molecular weight dependency might be over interpreted, as a molecule with a MW of 1000 would already possess sufficient binding sites to cover the TMD surface. Other reasons such as viscosity of the liquid etc. might play an equally important role so that care should be taken with the data interpretation.
- The authors should attempt to quantify the monolayer content. This might be challenging in AFM due to adsorbed dextran, but is actually important to benchmark the exfoliation efficiency compared to commonly used surfactants. Previous studies on WS₂ have suggested this can be done in a Raman/PL measurement or based on the A-exciton position (*ACS Nano*, 2016, 10 (1), pp 1589–1601).
- In context with the PL measurements, the authors speak of high intensities. Such statements should be avoided or quantified. High compared to what? It's very likely that the PL quantum yield is significantly below 1% and such PL should not be called intense. Unless the authors can quantify this (for example by actually measuring the quantum yield), these sections should be rephrased.
- L116: "glucose might be too weak to overcome the van der Waals interaction". This should be rephrased, as the energy to overcome the van der Waals interaction is supplied by the sonication and additives rather prevent the nanosheets from reaggregating, but are not necessarily responsible for the exfoliation.
- The Raman peaks of WS₂ were assigned as E_{2g} and A_{1g}. However, with green laser excitation, the main phonon is the 2LA(M) mode. This should be corrected.
- More details should be given in the method section such as: how were samples deposited for microscopic and spectroscopic (Raman, XPS, IR) characterisation? What were the g-forces in the centrifugation? How was unbound dextran removed?

Reviewer 1:

1. Is there any reference for “WS₂ mostly underwent intermolecular hydrogen bonding—rather than intramolecular hydrogen bonding”?

Response:

To the best of our knowledge, it is the first report to exfoliate and simultaneously functionalize TMD nanosheets with dextran through multivalent hydrogen bonding. Dextran itself can make intramolecular hydrogen bonding, which can be observed in its FT-IR spectrum showing that the C–O stretching modes appeared very broad at approximately 1004 cm⁻¹. However, the C–O stretching modes in dex-WS₂ nanosheets appeared sharp and were more clearly resolved at 1046, 1021, and 993 cm⁻¹, because the dextran on the basal plane of WS₂ mostly underwent intermolecular hydrogen bonding—rather than intramolecular hydrogen bonding—with the sulfur atoms. The characteristics of the vibration modes of dextran undergoing intermolecular or intramolecular hydrogen bonding were previously reported in the literature that we already cited [*Carbohydrate Research*, **2002**, 337(16): 1445-1451].

2. The first report on the incubation of TMD flakes on cells for their recognition is Nano Letters 14 (2), 857-863, 2014. Include and discuss it.

Response:

As the reviewer suggested, we have discussed the PL quenching of quasi-2D MoS₂ caused by K⁺ ions released from yeast cells with citation of the aforementioned paper in the revised manuscript as shown below.

On page 15:

“Notably, the Raman signals (2LA(M) and A_{1g}) of dex-WS₂ remained very intense with almost the same intensity even after binding on E. coli whereas its fluorescence was completely quenched (Supplementary Fig. 11). This fluorescence quenching of dex-WS₂ might be caused by the energy transfer to a variety of proteins on E. coli or the charge transfer to K⁺ ions released out of the bacteria as previously observed in quasi-2D MoS₂ adsorbed on yeast cells [*Nano Letter*, **2014**, 14(2), 857-863].”

3. What are the lateral dimensions of the flakes. Add it to figure 1. Is there any significance on the lateral dimension in terms of corners and edges effects in your work.

Response:

As shown in Supplementary Fig. 2, the average lateral size of dex-WS₂, dex-WSe₂, and dex-MoSe₂ was 48.96, 54.65, and 54.32 nm, respectively.

As discussed in the previous literature [*Nat. Commun.*, **2014**, 5, 4576; *ACS Nano*, **2016**, 10, 1589-1601], the extinction coefficient of WS₂ at 235 nm is independent on its lateral size whereas the extinction at 297 nm is dependent on it. To confirm the edge and confinement effects, the dex-WS₂ nanosheets were obtained at various centrifugation rates. Then, after quantification of their concentration using ICP-AES, their extinction spectra were measured as shown below. The extinction at 235 nm was almost same for all the dex-WS₂ obtained at the different centrifugation rates while the extinction at 290 nm decreased with increasing the centrifugation rate. The A-exciton absorption also blue-shifted as the centrifugation rate increased. This result reveals that the edge and confinement effects are valid in the dex-WS₂ nanosheets.

We then estimated the lateral size of dex-WS₂ nanosheets obtained at a centrifugation rate of 5000-10,000 rpm using the ratio between the extinction intensities at 235 and 290 nm as previously reported in the literature.

$$L(\text{nm}) = \frac{2.3 - Ext_{235}/Ext_{290}}{0.02Ext_{235}/Ext_{290} - 0.0185}$$

Where L is an estimated lateral size, Ext₂₃₅ and Ext₂₉₀ are the extinction intensities at 235 and 290 nm, respectively. The estimated lateral size of the dex-WS₂ nanosheets was found to be 49.64 nm, which is very similar with their average size obtained by TEM (48.96 nm).

We have moved the size histograms of dex-WS₂, dex-WSe₂, and dex-MoSe₂ from the Supplementary Information to Fig. 1 as the reviewer suggested. In addition, we have added more discussion about edge and confinement effects in the revised manuscript as shown below.

On page 7:

“To confirm edge and confinement effects on the absorption of dex-WS₂ [*Nat. Commun.*, **2014**, 5, 4576; *ACS Nano*, **2016**, 10, 1589-1601], dex-WS₂ was obtained at various centrifugation rates, and then its absorption was taken (Supplementary Fig. 6). After normalization of the absorption spectra with the concentration of dex-WS₂ measured by ICP-AES the absorbance at 235 nm and 290 nm was plotted as a function of centrifugation rates. The absorbance at 235 nm was independent on the centrifugation rate whereas the absorbance at 290 nm decreased with increasing the centrifugation rate. It was also found that the wavelength of the A excitonic absorption of dex-WS₂ blue-shifted as the centrifugation rate increased. This result reveals that the edge and confinement effects are valid for dex-WS₂.”

Figure 1. Exfoliation and functionalization of TMDs with dextran in an aqueous solution via multivalent hydrogen bonding. a) Schematic illustration of the exfoliation and functionalization of TMDs via multivalent hydrogen bonding. Photographs of the solution of TMDs exfoliated b) without dextran, c) with dextran, d) with glucose, and e) with PEG. TEM images of f) dex-WS₂, g) dex-WSe₂, and h) dex-MoSe₂ (scale bar, 50 nm). Height and lateral

size profiles of i) dex-WS₂, j) dex-WSe₂, and k) dex-MoSe₂ obtained using AFM.

Supplementary Figure 6. Dependence of the extinction property of dex-WS₂ on centrifugation rates. a) Extinction spectra of dex-WS₂ obtained at various centrifugation rates. b) Plot of extinction coefficients at 235 and 290 nm against centrifugation rates. The mean value of the extinction coefficient at 235 nm is 93.45 L g⁻¹cm⁻¹. All error bars represent a standard deviation from the mean values (n = 4).

4. I am not quite convinced that the reason for Raman signal selectivity for the last figure is what you presented. Are you able to add one more evidence (image, FTIR discussion, ...). The selectivity is remarkable though.

Response:

As the reviewer suggested, we have collected more evidences to show that dex-WS₂ is able to selectively recognize *E. coli* O157:H7 against *Salmonella* and *S. aureus*. After incubation of each bacteria with dex-WS₂ (25 μg mL⁻¹) for 1 h, the bacteria was washed with PBS several times to remove unbound dex-WS₂. Then, after centrifugation at 3000 rpm, the optical photograph of a pile of bacteria was taken as shown in Supplementary Fig. 10a. *E. coli* O157:H7 itself has a white color before treatment with dex-WS₂. However, after reaction with dex-WS₂, the color of *E. coli* O157:H7 turned light yellow, indicating that dex-WS₂ bound on the bacteria. In contrast, *Salmonella* and *S. aureus* retained a white color even after reaction with dex-WS₂, indicating that dex-WS₂ hardly bound on these two bacteria. This result clearly reveals that dex-WS₂ is able to selectively recognize *E. coli* O157:H7 without the use of antibodies.

We further analyzed the concentration of the dex-WS₂ bound on the bacteria using ICP-AES. As shown in Supplementary Fig. 10b, 6.95 μg mL⁻¹ of WS₂ was detected from the *E. coli* O157:H7 treated with dex-WS₂ whereas no noticeable WS₂ was measured from the *Salmonella* and *S. aureus* treated with dex-WS₂. This result verifies again that dex-WS₂ is able to selectively recognize *E. coli* O157:H7 against *Salmonella* and *S. aureus*.

We have added these new evidences for the selective recognition of *E. coli* O157:H7 by dex-WS₂ in the Supplementary Information, and have discussed it in the revised manuscript as shown below.

On page 14:

“After washing each bacteria with phosphate buffered saline (PBS) to remove unbound dex-WS₂, the optical photograph of a pile of bacteria centrifuged was taken (Supplementary Fig. 10a). The dex-WS₂-treated *E. coli* turned light yellow from white after reaction with dex-WS₂, indicating that dex-WS₂ bound on the surface of the bacteria. However, *Salmonella*

typhimurium and *S. aureus* retained an original white color even after reaction with dex-WS₂, indicating that dex-WS₂ hardly bound on these bacteria.”

“The content of dex-WS₂ bound on the bacteria was further quantified using ICP-AES (Supplementary Fig. 10b). A large amount of WS₂ (6.95 $\mu\text{g mL}^{-1}$) was measured from the dex-WS₂-treated *E. coli* whereas a trace amount of WS₂ was observed from the dex-WS₂-treated *Salmonella typhimurium* and *S. aureus*. This is in good agreement with the result of the bacteria detection based on the Raman signal measurement on glass microarrays. This result reveals again that dex-WS₂ was able to selectively recognize *E. coli* without the use of antibodies.”

Supplementary Figure 10. Additional confirmation of the selective binding of dex-WS₂ on *E. coli* O157:H7. a) Optical photograph of a pile of bacteria before and after reaction with dex-WS₂. b) Concentration of WS₂ bound on the dex-WS₂-treated bacteria using ICP-AES. All error bars represent a standard deviation from the mean values ($n = 4$).

Reviewer 2:

1. According to the results demonstrated by the authors, the functionalization of the TMDs with dextran is a natural process of the exfoliation, it is not a designed functionalization process which claimed by the authors in Introduction, “a simultaneous exfoliation and functionalization strategy” (Page 3 Line 63). The application example is to use that nature of such a functionalized surface to capture *E. coli*. Thus, the application of the materials is limited. Unless the authors could demonstrate other applications/or exfoliation/functionalization example based on this principle?

Response:

A simultaneous TMD exfoliation and functionalization strategy for the selective recognition of *E. coli* O157:H7 is a rationally designed process. In order to verify that this principle works for other systems, we have demonstrated one more example as a review-only material shown below.

For the selective recognition of glycosylated hemoglobin (HbA1c) for diagnosis of diabetes, boronic acid-modified poly(vinyl alcohol) (B-PVA) was synthesized and employed as a molecule for the simultaneous exfoliation and functionalization of WS₂ nanosheets in an aqueous solution as shown below (Figure R1). B-PVA-exfoliated and functionalized WS₂ nanosheets (B-PVA-WS₂) exhibited characteristic photoluminescence (PL) at 620 nm corresponding to a direct band gap of WS₂ monolayers. The PL of B-PVA-WS₂, however, was quenched in the presence of HbA1c, and the quenching response gradually increased with increasing the concentration of the target (Figure R1b), indicating that B-PVA-WS₂ is able to recognize HbA1c. In contrast, the WS₂ nanosheets exfoliated by PVA not bearing boronic acid (PVA-WS₂) showed no PL quenching in the presence of HbA1c (Figure R1c).

These results clearly reveal that our simultaneous TMD exfoliation and functionalization strategy is a rationally designed process that is applicable for various target systems of interest. The results for the selective recognition of HbA1c by B-PVA-WS₂ are now under review for publication in the other journal.

[Redacted]

2. For the bacteria detection, it is also important to differentiate E. coli O157:H7 from other E.coli as well. However, other E.coli may also contains fimH protein on their membrane (see Nature Communication, 2016, 7, 10738).

Response:

As the reviewer suggested, we have further investigated for the specificity of dex-WS₂ for

the detection of *E. coli* O157:H7 against other strains of *E. coli* such as *E. coli* K-12 and *E. coli* O1:K1:H7. They were incubated with dex-WS₂ under the same condition as *E. coli* O157:H7. After washing out unbound dex-WS₂ from the *E. coli* K-12 and *E. coli* O1:K1:H7 solutions, the Raman signals of dex-WS₂ at 352 cm⁻¹ were collected from the dex-WS₂-treated bacteria on microarrays. As shown in Supplementary Fig. 12, the strong Raman signals of dex-WS₂ were observed only from *E. coli* O157:H7. The very weak Raman signals were measured from both *E. coli* K-12 and *E. coli* O1:K1:H7. This result clearly reveals that dex-WS₂ is able to specifically recognize *E. coli* O157:H7 against different strains of *E. coli*.

We speculate that the specificity of dex-WS₂ to *E. coli* O157:H7 might arise from a structural difference in the *fimH* of *E. coli* or the recognition capability of the 3D structure of dextran on the rigid surface of WS₂ nanosheets. Further investigations for understanding the specificity of dex-WS₂ to *E. coli* O157:H7 will be carried out in the follow-up paper.

We have added this new result in the Supplementary Information and have discussed about the specificity of dex-WS₂ to *E. coli* O157:H7 with citation of the above mentioned paper in the revised manuscript as shown below.

On page 15:

“To investigate the specificity of dex-WS₂ to *E. coli* O157:H7, other strains of *E. coli* such as *E. coli* K-12 and *E. coli* O1:K1:H7 were incubated with dex-WS₂, and then the 2LA(M) Raman signals at 352 cm⁻¹ were collected from the bacteria on glass microarrays. As shown in Supplementary Fig. 12, the strong Raman signals of dex-WS₂ were observed only from *E. coli* O157:H7. The very weak Raman signals of dex-WS₂ were measured from both *E. coli* K-12 and *E. coli* O1:K1:H7. This result clearly reveals that dex-WS₂ was able to specifically recognize *E. coli* O157:H7 against other strains of *E. coli*. It is worth noting that dex-WS₂ could differentiate *E. coli* O157:H7 from other strains of *E. coli* K-12 and *E. coli* O1:K1:H7 although they all have *fimH* on their membrane [*Nature Communication*, **2016**, 7, 10738; *Nat. Rev. Microbiol.*, **2009**, 7, 765–774]. We speculate that the recognition specificity of dex-WS₂ might be ascribed to the fact for the structural difference of the *fimH* on the *E. coli* or the specific recognition capability of a three-dimensional structure of dextran adsorbed on the rigid surface of WS₂ nanosheets.”

Supplementary Figure 12. Recognition Specificity of dex-WS₂ to *E. coli* O157:H7 against different strains of *E. coli*. a) Raman images of the dex-WS₂-treated bacteria based on the 2LA(M) peak of dex-WS₂ at 352 cm⁻¹, showing the selective recognition of *E. coli* O157:H7 against *E. coli* K-12 and *E. coli* O1:K1:H7. b) The normalized total intensity of the 2LA(M) Raman signal of dex-WS₂ bound on three different strains of *E. coli*. All error bars represent a standard deviation from the mean values (n = 4).

3. Using dextran based polymer to exfoliate 2D materials has been reported before, see *Macromolecules*. 2015, Vol.48(18), p.6628-6637.

Response:

The *Macromolecules* paper [*Macromolecules*, **2015**, 48(18), 6628-6637], which the reviewer mentioned above, is not the one demonstrating the exfoliation and functionalization of TMDs with dextran in an aqueous solution. This *Macromolecules* paper reported the exfoliation of **graphene** using water-soluble polymers such as hydrophobic group-modified dextran and PVA. What this reference found is that the hydrophobic interaction of the polymers with graphene was essential for the effective exfoliation of hydrophobic graphene in water.

However, in our current work, we have demonstrated the simultaneous exfoliation and functionalization of TMD nanosheets such as WS₂, WSe₂, and MoSe₂ with dextran via multivalent hydrogen bonding. We have also verified that the multivalent hydrogen bonding of dextran was a primary factor for the effective exfoliation and functionalization of TMD nanosheets in water via several experiments and DFT simulations. In particular, as-prepared dex-WS₂ was able to recognize *E. coli* O157:H7 in a selective and specific manner without the use of any antibodies.

We believe what we report in our paper is obviously novel and includes a lot of new scientific results and significance in relation to Nanoscience, Nanobio Technology, Biosensing, and Materials Science.

4. The authors using the UV-Vis absorbance to compare the amount of nanosheet extracted. Though in general this may indirect reflect the amount of nanosheet extracted, but the optical property of nanosheet not only determined by the concentration, but also the size and shape of the nanosheet.

Response:

According to the previous papers [*Nat. Commun.*, **2014**, 5, 4576; *ACS Nano*, **2016**, 10, 1589-1601; *Adv. Funct. Mater.*, **2016**, 26(7): 1028-1039], the absorption of TMDs in a UV region was found to be dependent only on their concentration, but not dependent on their size and thickness. However, the A-excitonic absorption of TMDs in a visible region was influenced by their structures such as size and thickness. Hence, the UV absorbance can be used to find out a tendency in the exfoliation efficiency of dex-TMD nanosheets.

In order to confirm that the UV absorbance-based quantification of TMD nanosheets is reliable, the concentration of TMD nanosheets exfoliated at the various ratios of bulk WS₂ with dextran was measured by ICP-AES. As shown in Supplementary Fig. 1c, the tendency of the exfoliation efficiency as a function of WS₂-to-dextran ratios is in good agreement with the UV absorbance-based plot (Supplementary Fig. 1b). According to the ICP-AES results, the exfoliation efficiency of dex-WS₂ was still highest at a ratio of 15:1 (bulk WS₂: dextran).

We have added this new ICP-AES result in the Supplementary Information, and have discussed about this quantification result in the revised manuscript as shown below.

On page 6:

“The amount of dex-WS₂ nanosheets was additionally quantified by inductively coupled plasma-atomic emission spectroscopy (ICP-AES) (Supplementary Fig. 1c). The tendency of the exfoliation efficiency measured by ICP-AES over the weight ratios was in good agreement with the UV absorbance-based plot. The highest concentration of dex-WS₂ nanosheets was obtained at a ratio of 15:1.”

Supplementary Figure 1. Effect of the weight ratio of bulk WS₂ with dextran on the exfoliation efficiency of dex-WS₂. a) Optical photographs of the solution of dex-WS₂ nanosheets exfoliated at the various ratios of bulk WS₂ with dextran. b) Plot of the absorbance (297 nm) of the dex-WS₂ solutions with the ratios of bulk WS₂ with dextran. c) ICP-AES-measured concentrations of the dex-WS₂ nanosheets exfoliated at the different ratios of bulk WS₂ with dextran. All error bars represent a standard deviation from the mean values (n = 4).

5. The authors claimed that the dex-TMDs have a better affinity to *E. Coli* O157:H7 than *E. Coli*-specified antibodies and aptamers. They referred to Refs. 45 & 46. It is

questionable whether such a comparison is fair or not since the experiments conditions are very different. Such a claim can be concluded if the authors can design a convincing comparison experiment by considering the relative concentration of the capturing agents, the same bacteria, under the same operation conditions.

Response:

As the reviewer know, the affinity of molecules to a target of interest depends on their physical and chemical properties that can be affected by environmental conditions. Hence, most of the affinities of the molecules reported in the literature were measured in the optimal conditions where the highest affinity can be provided.

The affinity of the *E. coli* O157:H7-specific antibody, which was compared with that of dex-WS₂, was also measured in the optimal conditions, and its highest value was reported in the literature. According to the previous result in the reference, the affinity of the *E. coli* O157:H7-specific antibody to the bacteria noticeably decreased as the pH value of the solution changed from pH 5.0 to pH 7.0. As the affinity of dex-WS₂ to *E. coli* O157:H7 was measured at pH 7.4, we found that the affinity of the *E. coli* O157:H7-specific antibody at pH 7.0 was much lower than its highest value measured at pH 5.0 as well as the affinity of dex-WS₂.

Hence, we believe that it is more reasonable to compare the highest affinities of both dex-WS₂ and the *E. coli* O157:H7-specific antibody obtained at their own optimal conditions since the affinity of the antibody will decrease with a change at the operation conditions.

However, to clarify this issue more clearly, we have measured the affinity of dex-WS₂ and the *E. coli* O157:H7-specific antibody against the bacteria *E. coli* O157:H7 using a quartz crystal microbalance (QCM) instrument (QSence-E4). After self-assembling 11-mercaptopundecanoic acid (MUA, 10 mM) and 11-hydroxy-1-undecanethiol (HUT, 50 mM) for 8 h at 40 °C on a gold QCM chip, the carboxyl group of MUA on the chip was activated by EDC/NHS (10 mM EDC and 20 mM NHS) for 1 h. Then, an antimicrobial peptide (KNYSSSISSIHIC) (20 mL, 0.1 mg·mL⁻¹) was immobilized for 24 h to capture the *E. coli* on the QCM chip. After washing the chip with DI water and EtOH, the bacteria solution (5 × 10⁶ CFU mL⁻¹) flowed into the chip with a rate of 50 μL min⁻¹ for 30 min. After washing it with PBS for 30 min, dex-WS₂ or the antibody flowed into the QCM chip by changing its concentration from 2 to 16 nM for 30 min. Finally, the chip was washed with PBS for 30 min.

As shown in Figure R2, the affinity of dex-WS₂ to *E. coli* O157:H7 was still higher than

that of the antibody when it was measured at the same conditions.

Nevertheless, we still believe that it is more reasonable to compare the highest affinity of the *E. coli* O157:H7-specific antibody obtained at its own optimal conditions with that of dex-WS₂.

Figure R2. Measurement of the affinity of dex-WS₂ and the *E. coli* O157:H7-specific antibody against *E. coli* O157:H7 using a quartz crystal microbalance (QCM) instrument.

Reviewer 3:

1. The authors discuss that defects on the TMDs may further help the binding. While this seems valid from the theoretical viewpoint, it is nonetheless a bit worrying that the oxide content according to XPS is usually high. The authors should test whether these defects are introduced on exfoliation via sonication or already present in the starting material. If they are introduced by the sonication, the authors should minimize this impact for example by lowering the amplitude to avoid damage to the sheets. Otherwise, the question remains whether it would also work for defect-free nanosheets.

Response:

As the reviewer suggested, the oxide content of bulk WS₂ and fresh dex-WS₂ was measured again by XPS. As shown Supplementary Fig. 9, the starting bulk WS₂ already has a 15% portion of WO₃. After exfoliation of the bulk WS₂ into dex-WS₂ nanosheets, the proportion of WO₃ in the dex-WS₂ nanosheets increased to 31%, indicating that oxidation occurred during a process of exfoliation. In order to minimize the oxidation of WS₂ nanosheets, the sonication amplitude was reduced from 75% to 37.5%. As shown in Supplementary Fig. 9h, the proportion of WO₃ in the dex-WS₂ nanosheets significantly decreased to 21% along with a decrease in the exfoliation yield of the dex-WS₂ nanosheets.

We have added the XPS spectrum of bulk WS₂ in the Supplementary Information, and have discussed about the oxidation of dex-WS₂ nanosheets during an exfoliation process in the revised manuscript as shown below.

On page 11:

“In the X-ray photoelectron spectroscopy (XPS) spectra of dex-WS₂ (Supplementary Fig. 9), the characteristic peaks of dextran were also observed, demonstrating that dex-WS₂ hybrids were successfully prepared. Moreover, the W4f peaks of dex-WS₂ confirmed that it retained an intrinsic 2H phase after exfoliation and functionalization. It was observed that dex-WS₂ was partially oxidized to produce tungsten oxide during a process of exfoliation. The tungsten oxide content in dex-WS₂ increased to 31% as compared to the proportion (15%) of starting bulk WS₂. However, the tungsten oxide content in dex-WS₂ significantly decreased to 21% as the

sonication amplitude was reduced to a half.”

Supplementary Figure 9. XPS analysis of dextran, bulk WS₂, and dex-WS₂. a) C1s, and b) O1s spectra of dextran. c) C1s, d) O1s, e) W4f, and f) S2p spectra of dex-WS₂ nanosheets. g) W4f spectrum of bulk WS₂. h) W4f spectrum of dex-WS₂ exfoliated at a lower amplitude (37.5%).

2. The molecular weight dependency might be over interpreted, as a molecule with a MW of 1000 would already possess sufficient binding sites to cover the TMD surface. Other reasons such as viscosity of the liquid etc. might play an equally important role so that care should be taken with the data interpretation.

Response:

We have measured the viscosity of the aqueous solution containing the dextran of MW 40,000 or the one of MW 1000. However, both the dextran-containing aqueous solutions have the same viscosity (20.8 cP) as pure water. The amount of dextran (0.2 wt% to water) was too small to influence the viscosity of water. This result clearly reveals that the viscosity of an aqueous solution is not a factor to affect the exfoliation efficiency of TMD nanosheets.

As discussed, the number of multivalent hydrogen bonding between TMD nanosheets and dextran is a crucial factor for the effective exfoliation and functionalization of TMD nanosheets in an aqueous solution. The DFT simulation results also suggest that the multivalent hydrogen bonding plays an important role for the simultaneous exfoliation and functionalization of TMD nanosheets with dextran in an aqueous solution.

3. The authors should attempt to quantify the monolayer content. This might be challenging in AFM due to adsorbed dextran, but is actually important to benchmark the exfoliation efficiency compared to commonly used surfactants. Previous studies on WS₂ have suggested this can be done in a Raman/PL measurement or based on the A-exciton position (ACS Nano, 2016, 10 (1), pp 1589–1601).

Response:

As the reviewer pointed out, it is very challenging to quantify the accurate content of TMD monolayers in the dex-TMD solution using AFM due to adsorbed dextran. As reported in the literature [ACS Nano, 2016, 10 (1), 1589–1601], the optical properties of TMD nanosheets could be used for the quantification of a monolayer content in an exfoliated TMD solution. In particular, a PL/Raman intensity ratio was found to be quantitatively linked with the monolayer content.

Therefore, we have also tried to quantify the monolayer content of dex-WS₂ by applying a PL/Raman intensity ratio into the equation shown below. To find out if this optical method for the quantification of a monolayer content is also valid for dex-TMDs, the plot of a PL/Raman intensity ratio versus a centrifugation g-force for dex-WS₂ was first obtained. To this end, the dex-WS₂ solutions were collected at various centrifugation rates and their PL and Raman spectra were taken as reported in the reference [ACS Nano, 2016, 10 (1), 1589–1601]. As shown in Figure R3 below, the PL/Raman intensity ratio of dex-WS₂ increased with increasing

the centrifugation g-force, indicating that the monolayer content gradually increased. However, the slope of the plot was so different from the one reported in the aforementioned reference, indicating that the previously-reported metric for the quantification of the monolayer content could not give the accurate monolayer content for dex-WS₂.

$$V_f = \frac{1}{17} \frac{I_{PL}}{I_{Raman}}$$

The PL/Raman intensity ratio of dex-WS₂ obtained at the centrifugation rate of 5000-10,000 rpm was 0.9141, which was then applied to the previously-reported metric shown below to quantify the monolayer content [ACS Nano, 2016, 10 (1), 1589–1601]. The estimated monolayer volume fraction in dex-WS₂ was 5.4%. However, this estimated monolayer content is much lower than what we expected based on the results of the AFM height profile and the optical properties of dex-WS₂. As we expected in the plot of a PL/Raman intensity ratio versus a centrifugation g-force for dex-WS₂, this metric could not give the accurate monolayer content of the dex-WS₂. Although AFM does not provide the accurate fraction of WS₂ monolayers due to dextran adsorption, it could give the information for the minimum monolayer content. As shown in the height profile of dex-WS₂ (Fig. 1i), the minimum monolayer content was at least 25%. In addition, no other characteristic PL and Raman peaks for bi- and tri-layers of WS₂ were observed in the spectra of dex-WS₂. The PL and Raman properties clearly suggest that a majority of dex-WS₂ nanosheets were monolayers.

There are two expected reasons why the previously-reported metric based on a PL/Raman intensity ratio for the quantification of the monolayer content was not applicable for dex-WS₂. First, the PL intensity of WS₂ nanosheets can be affected by the type of a wrapping molecule while the Raman intensity is much less susceptible to the effect of the wrapping compound. Therefore, the quantum yield (QY) of dex-WS₂ might be so different from the QY of surfactant-exfoliated WS₂. Second, the QY of exfoliated WS₂ nanosheets can also be influenced by the extent of defects. As discussed, the oxidation of dex-WS₂ took place during a process of exfoliation, which can affect the QY of the WS₂ nanosheets. Non-radiative recombination can occur at these defect sites, resulting in a decrease in the QY.

To clarify this issue, we have more discussed about the monolayer content of dex-WS₂ in the revised manuscript as shown below.

On page 9:

“As reported in the previous literature [ACS Nano, 2016, 10 (1), 1589–1601], we tried to estimate the monolayer volume fraction of dex-WS₂ by applying its PL/Raman intensity ratio to the reported metric. However, the estimated monolayer volume fraction of dex-WS₂ was much lower than what expected based on its AFM height profile and PL/Raman properties. As discussed in the PL and Raman spectra of dex-WS₂ above, a majority of dex-WS₂ should be monolayers. There are two expected reasons why the previous metric based on a PL/Raman intensity ratio for monolayer volume fractions was not valid for dex-WS₂. First, the PL intensity of WS₂ nanosheets can be affected by an exfoliating molecule. The quantum yield (QY) of dex-WS₂ might be different from that of surfactant-exfoliated WS₂ nanosheets. Second, the QY of WS₂ nanosheets can be influenced by the extent of defects in which non-radiative recombination takes place, resulting in a decrease in the QY.”

Figure R3. Plot of a PL/Raman ratio against a centrifugation g-force for dex-WS₂ obtained at various centrifugation rates.

4. In context with the PL measurements, the authors speak of high intensities. Such statements should be avoided or quantified. High compared to what? It’s very likely that the PL quantum yield is significantly below 1% and such PL should not be called intense.

Unless the authors can quantify this (for example by actually measuring the quantum yield), these sections should be rephrased.

Response:

To clarify this issue, we have rephrased the sentence for the PL intensity of dex-TMDs in the revised manuscript as shown below. Figure R4 shows that the PL of dex-WS₂ was much more intense than that of bulk WS₂.

On page 8:

“As compared to bulk WS₂, strong PL emission of dex-WS₂ appeared at 612 nm (black line, Fig. 2c), showing the transformation of an indirect semiconductor of bulk WS₂ into a direct semiconductor of WS₂ monolayer.”

Figure R4. Fluorescence spectra of dex-WS₂ and bulk WS₂ under excitation at 532 nm.

5. L116: “glucose might be too weak to overcome the van der Waals interaction”. This should be rephrased, as the energy to overcome the van der walls interaction is supplied by the sonication and additives rather prevent the nanosheets from reaggregating, but

are not necessarily responsible for the exfoliation.

Response:

As the reviewer suggested, the above sentence has been re-phrased in the revised manuscript as shown below.

On page 6:

“As shown in Fig. 1d, no TMD nanosheets were obtained in the presence of glucose. As the hydrogen binding multivalency of glucose is two hundred times smaller than that of dextran, the interaction of TMDs with glucose might be too weak to overcome the van der Waals interaction between the TMD interlayers for re-stacking.”

6. The Raman peaks of WS₂ were assigned as E_{2g} and A_{1g}. However, with green laser excitation, the main phonon is the 2LA(M) mode. This should be corrected.

Response:

As the reviewer pointed out, the vibrational modes, 2LA(M) and E¹_{2g}, of WS₂ nanosheets appear overlapped around at 350 cm⁻¹. However, the intensity of the 2LA(M) of the WS₂ nanosheets is more intense than that of the E¹_{2g} under excitation at 532 nm [*Sci. Rep.*, **2013**, 3, 1755].

We have revised the peak assignment of the Raman modes of dex-WS₂ in the revised manuscript as shown below.

On page 8:

“In the Raman spectra of dex-WS₂ (black line), two distinct peaks for a combination of in-plane vibrational mode (E¹_{2g}) and longitudinal acoustic mode (2LA(M)), and an out-of-plane vibrational mode (A_{1g}) appeared at 350 and 418 cm⁻¹, respectively [*Sci. Rep.*, **2013**, 3, 1755]. As the A_{1g} mode of WS₂ is more significantly dependent on its layer number than the 2LA(M) mode, their intensity ratio can be used as an indicator to determine the number of WS₂ layers [*Sci. Rep.*, **2013**, 3, 1608]. The intensity ratio of the A_{1g} peak to the 2LA(M) peak in the Raman spectrum of dex-WS₂ was 6.7, indicating that monolayer dex-WS₂ was mainly exfoliated by

dextran.”

7. More details should be given in the method section such as: how were samples deposited for microscopic and spectroscopic (Raman, XPS, IR) characterization? What were the g-forces in the centrifugation? How was unbound dextran removed?

Response:

We have revised the methods to give more details of the procedures for the preparation and characterization of dex-TMDs in the revised manuscript as shown below.

On page 17:

Synthesis of TMD nanosheets with glucan multivalency

In this study, dex-TMD nanosheets were obtained via liquid exfoliation of bulk TMDs in the presence of dextran (MW 40,000) in water. Accordingly, 6 g of bulk TMD (8.4 g for WSe₂) was added to 200 mL of water containing dextran (2 g L⁻¹), followed by sonication using a tip sonicator in an ice bath for 5 h at 75% amplitude with a pulse of 6s-on and 2s-off. The resulting solution was centrifuged at 3000 rpm (1977 g-force) for 1.5 h to discard unexfoliated TMDs. The supernatant was collected and centrifuged again at 10,000 rpm (15,344 g-force) for 1.5 h. The sediment was collected, but the supernatant was discarded to remove unbound dextran and a smaller size of TMD particles. Subsequently, the sediment obtained at 10,000 rpm was re-dispersed in 25 mL of water and centrifuged again at 5000 rpm (3024 g-force) for 1.5 h. The supernatant solution was finally collected to obtain dex-TMD monolayers (5000–10,000 rpm).

On page 2 in the Supplementary Information:

Characterization of dex-TMDs

For analysis of dex-TMDs with Raman and XPS, 10 μL of as-prepared dex-TMDs were dropped on a Si wafer and dried at room temperature for 4 h. Then, the Raman or XPS spectra of dex-TMDs were measured. The PL of dex-TMDs were measured under excitation at 532 nm (9 W). For FT-IR analysis, dex-TMDs were lyophilized for 24 h, and then the FT-IR spectra of dex-TMDs (10 mg) were obtained using an ATR module. To obtain the TEM images of dex-

TMDs, 100 μL of the dex-TMDs solution was diluted with 10 mL of water. A 10 μL of the resulting solution was dropped on a Cu grid, followed by drying for 12 h at room temperature.

Reviewers' comments:

Reviewer #1 (Remarks to the Author):

The authors have comprehensively addressed the comments by the three reviewers

Reviewer #3 (Remarks to the Author):

The authors have thoroughly revised their manuscript, including new experimental data and modified the manuscript and SI accordingly. The manuscript is now suitable for publication except for a final small remark:

The authors measured and included extinction spectra and their interpretation with reference to literature on this subject. However, they refer to them as absorption spectra which is not correct, as they clearly include a non-resonant scattering background typical for nanosheets. Absorption spectra on nanomaterials can only be obtained when scattered light is collected by an integrating sphere. This should be corrected.

Reviewer #4 (Remarks to the Author):

1. According to the results demonstrated by the authors, the functionalization of the TMDs with dextran is a natural process of the exfoliation, it is not a designed functionalization process which claimed by the authors in Introduction, "a simultaneous exfoliation and functionalization strategy" (Page 3 Line 63). The application example is to use that nature of such a functionalized surface to capture E. coli. Thus, the application of the materials is limited. Unless the authors could demonstrate other applications/or exfoliation/functionalization example based on this principle?

In response to 1. the Author has attempted to explain the rationale behind their TMDs exfoliation and functionalisation strategy using dextran, with the exfoliation and functionalisation of WS2 nanosheets using boronic acid modified poly (vinyl alcohol) (B-PVA). Dextran and B-PVA are substantially different systems with diverse chemical and physical properties, and it is not clear what would be the common ground for a rationally designed approach for the TMDs' exfoliation and functionalisation. The Author has claimed in the main body of the paper that the simultaneous exfoliation and functionalisation of TMDs nanosheets is enabled by the cluster glycoside effect of dextran (page 6, line 120), which has no pertinence or relation with the B-PVA system they proposed as a demonstration of their rationally designed approach.

I'd like also to clarify that the cluster glycoside effect is a very fascinating but complex and concerted carbohydrate-proteins interaction that requires clustered carbohydrate binding sites and a multivalent ligand that can present carbohydrates with proper orientation and spacing (Acc. Chem. Res., Vol. 28, 8, 1995). In this regard if the Author claims that the cluster glycoside effect of dextran is responsible for the functionalisation of TMDs, what then would be the mechanism responsible for the detection of E. Coli? If it's the same cluster glycoside effect that enabled the exfoliation and functionalisation, are there enough binding sites available for the claimed specific detection?

Although the effectiveness of the functionalisation with dextran has been quite extensively explained by the Author, it is not clear the rationale behind this approach, particularly when they have used a glucose-based system instead of a mannose-based system, the latter being three orders of magnitude more affine for the detection of E. Coli.

2. For the bacteria detection, it is also important to differentiate E. coli O157:H7 from other E. coli as well. However, other E. coli may also contains fimH protein on their membrane (see Nature Communication, 2016, 7, 10738).

In response to 2. The Author has shown that the Raman signal for the dex-WS2 E. coli O157:H7 is much stronger than the one observed for the E. coli K12 and E. coli O1:K1:H7. It is unclear and not sufficiently explained by the Author on which basis there is a preferential detection of E. coli O157:H7 opposed to the E. coli K12 and O1:K1:H7 strains, since they all have the same FimH protein. I would be very careful in claiming specificity of detection towards a particular strain since in literature there is still a wide debate on the binding mechanisms involved in the detection process.

I'd also ask the Author to review the use of the terms "specific" and "selective", since they have two very different meanings in analytical chemistry but the Author is using them indistinctively (see page 5 line 93 and line 96; caption in supplementary figure 12, etc.). According to the IUPAC definition (Pure Appl. Chem., Vol. 73, No. 8, pp. 1381–1386, 2001) "A specific reaction or test is one that occurs only with the substance of interest, while a selective reaction or test is one that can occur with other substances but exhibits a degree of preference for the substance of interest." The Author has not demonstrated specificity with the dex-WS2 system, since the Author has not shown comparison with other glycan-based systems for E. coli detection (for example mannose). Therefore, I would suggest replacing the term with something more appropriate such as "increased binding affinity" towards a particular strain.

3. Using dextran based polymer to exfoliate 2D materials has been reported before, see *Macromolecules*. 2015, Vol.48(18), p.6628-6637.

In response to 3. Please revise your use of the terms "specific" and "selective" they are not interchangeable. Although the report is showing an effective exfoliation and functionalisation of TMDs with dextran via multiple hydrogen bonding, this contribution has provided insufficient insights and inadequate rationale and experimental materials for what they claim to be a novel and significant report in relation to Biosensing.

5. The authors claimed that the dex-TMDs have a better affinity to E. coli O157:H7 than E. coli-specified antibodies and aptamers. They referred to Refs. 45 & 46. It is questionable whether such a comparison is fair or not since the experimental conditions are very different. Such a claim can be concluded if the authors can design a convincing comparison experiment by considering the relative concentration of the capturing agents, the same bacteria, under the same operation conditions.

In response to 5. The Author is comparing the affinity of E. coli O157:H7-specific antibody found in literature with that of dex-WS2, although the techniques used for the calibration curves are different: in the literature is used SPR and the Author is using QCM (as response to the issue raised in 5). The Author has compared the affinity of dex-WS2 with the E. coli O157:H7-specific antibody against the bacteria E. coli O157:H7 using QCM transducer platform, which could be used as a comparative technique to SPR if the exact conditions are met such as surface preparation and functionalisation process. In this case the surface functionalisation of the chip for the QCM is different from the one used in the literature, which could give a substantially different outcome. The author has anyway compared the binding affinity of E. coli O157:H7-specific antibody with dex-WS2 for the detection of E. coli O157:H7, resulting in higher affinity of dex-WS2. In the rebuttal is presented the graph of the $C/\Delta F$ vs concentration, it would be good to see also the original ΔF vs concentration. Also, there is no mention of a control experiment, with buffer for example, to have a measure of the signal to noise ratio, or with BSA for a non-specific binding control. It would be useful for determining the analytical figures of merit of the sensing platform to evaluate sensitivity and limit of detection.

Reviewer #3:

1. The authors measured and included extinction spectra and their interpretation with reference to literature on this subject. However, they refer to them as absorption spectra which is not correct, as they clearly include a non-resonant scattering background typical for nanosheets. Absorption spectra on nanomaterials can only be obtained when scattered light is collected by an integrating sphere. This should be corrected.

Response:

As the reviewer pointed out, it is more accurate to refer to what we measured as extinction spectra, although scattering is normally so weak in dilute solution.

Therefore, we have corrected absorption spectra with extinction spectra in the revised manuscript.

Reviewer #4:

1. In response to 1. the Author has attempted to explain the rationale behind their TMDs exfoliation and functionalisation strategy using dextran, with the exfoliation and functionalisation of WS₂ nanosheets using boronic acid modified poly (vinyl alcohol) (B-PVA). Dextran and B-PVA are substantially different systems with diverse chemical and physical properties, and it is not clear what would be the common ground for a rationally designed approach for the TMDs' exfoliation and functionalisation. The Author has claimed in the main body of the paper that the simultaneous exfoliation and functionalisation of TMDs nanosheets is enabled by the cluster glycoside effect of dextran (page 6, line 120), which has no pertinence or relation with the B-PVA system they proposed as a demonstration of their rationally designed approach.

I'd like also to clarify that the cluster glycoside effect is a very fascinating but complex and concerted carbohydrate-proteins interaction that requires clustered carbohydrate binding sites and a multivalent ligand that can present carbohydrates with proper orientation and spacing (Acc. Chem. Res., Vol. 28, 8, 1995). In this regard if the Author claims that the cluster glycoside effect of dextran is responsible for the functionalisation of TMDs, what then would be the mechanism responsible for the detection of E. Coli? If it's the same cluster glycoside effect that enabled the exfoliation and functionalisation, are there enough binding sites available for the claimed specific detection?

Although the effectiveness of the functionalisation with dextran has been quite extensively explained by the Author, it is not clear the rationale behind this approach, particularly when they have used a glucose-based system instead of a mannose-based system, the latter being three orders of magnitude more affine for the detection of *E. Coli*.

Response:

As emphasized in the manuscript, the primary factor responsible for the simultaneous exfoliation and functionalization of TMD nanosheets in an aqueous solution is *the multivalent hydrogen bonding of dextran* with the TMD surface rather than the cluster glycoside effect. As shown in Figure 1, glucose with a much smaller hydrogen bonding multivalency could not exfoliate and functionalize TMD nanosheets in the aqueous solution. In addition, when polyethylene glycol dimethyl ether (PEG), which cannot undergo hydrogen bonding with TMDs, was added during the exfoliation process, TMD nanosheets were hardly exfoliated in the aqueous solution. These experimental results clearly confirm that the *multivalent hydrogen bonding* of dextran with the TMD surface is essential for the simultaneous exfoliation and functionalization of TMD nanosheets in water.

As shown in Figure 3, dex-TMDs were found to be very stably dispersed in an aqueous solution even after successive centrifugation and re-dispersion. This result indicates that a certain portion of dextran chains would stretch toward the bulk solution to induce steric stabilization on TMD nanosheets during dispersion whereas the rest portion would stick to the surface of the TMD nanosheets. Therefore, we expect that dex-TMDs would still have enough binding sites for the selective recognition of *E. coli*, which has been clearly verified through the experimental results.

One more example, which can show the rationale behind the simultaneous exfoliation and functionalization of TMD nanosheets based on *multivalent hydrogen bonding*, is the WS₂ nanosheets exfoliated and functionalized by boronic acid-modified poly(vinyl alcohol) (B-PVA). As shown in Figure R1, B-PVA includes a lot of hydroxyl groups as well as boronic acid. The ratio of the hydroxyl groups with boronic acid on B-PVA was 13:1, which was quantified by measuring the absorbance of the boronic acid group. Therefore, B-PVA could undergo *multivalent hydrogen bonding* with the WS₂ surface during a sonication process to simultaneously exfoliate and functionalize the WS₂ nanosheets in an aqueous solution. This example reveals again that the *multivalent hydrogen bonding* of a polymer with the TMD

surface rather than the cluster glycoside effect is an essential factor for the effective exfoliation and functionalization of TMD nanosheets in water. As the boronic acid can recognize glycosylated hemoglobin (HbA1c), known as a biomarker for diagnosis of diabetes, the resulting B-PVA-exfoliated and functionalized WS₂ nanosheets (B-PVA-WS₂) were able to selectively detect HbA1c via quenching of their characteristic photoluminescence (PL) at 620 nm (Figure R1b). In contrast, the WS₂ nanosheets exfoliated by PVA not bearing boronic acid (PVA-WS₂) showed no PL quenching in the presence of HbA1c (Figure R1c).

This example of B-PVA-WS₂ clearly verifies that our strategy called “a simultaneous exfoliation and functionalization of TMDs based on *multivalent hydrogen bonding*” is a rational approach for designing functional TMD nanosheets with a selective recognition of target molecules of interest including *E. coli* O157:H7. In addition, this example obviously suggests that our approach can be extended to various TMD-based biosensors for the selective detection of targets of interest. The result of B-PVA-WS₂ has been accepted for publication in the other journal, and will be published soon.

Therefore, we have replaced “the cluster glycoside effect of dextran”, which was written only once in the main text, with “*the multivalent hydrogen bonding of dextran*” on page 6 in the revised manuscript.

[Redacted]

There are several reasons why we chose a glucose-based polymer, dextran, instead of a mannose-based polymer as an exfoliating and functionalizing molecule for the selective detection of *E. coli* O157:H7 although mannose is known to have a higher binding affinity to a *fimH* lectin ($K_d = 2.3 \mu\text{M}$) than glucose ($K_d = 9.4 \text{ mM}$). As described in the manuscript, we hypothesized that if TMDs would be functionalized with multivalent dextran (MW 40,000), an enhancement effect on the binding affinity of resulting dex-TMDs to *fimH*-containing bacteria

through a cluster glycoside effect and a newly-created 3D structure of dextran on the rigid surface of TMDs would be more distinctly observed as compared to glucose because glucose has a lower intrinsic affinity to *fimH* lectins. This enhanced binding affinity of dex-TMDs would eventually enable the selective detection of the pathogenic bacteria without the use of expensive antibodies or peptides. As we anticipated, the TMD nanosheets functionalized by dextran (dex-TMDs) exhibited a significantly-enhanced binding affinity to *E. coli* O157:H7, allowing the selective detection of *E. coli* against *Salmonella typhimurium* and *Staphylococcus* (*S.*) *aureus* (Figure 5 and 6). In addition, dex-TMDs exhibited a much higher binding affinity to *E. coli* O157:H7 as compared to different strains of *E. coli* bearing a slightly different component of *fimH* lectins such as *E. coli* K-12 and *E. coli* O1:K1:H7. As discussed, we speculate that the higher binding affinity of dex-TMDs to *E. coli* O157:H7 might arise from a slight structural difference in the *fimH* of *E. coli* or the newly-created recognition capability of the 3D structure of dextran on the rigid surface of the TMD nanosheets. If a mannose-based polymer was used for the exfoliation and functionalization of the TMD nanosheets, such a clear differentiation in the binding affinity against different strains of *E. coli* might be vague due to the strong intrinsic binding affinity of mannose to *fimH* lectins.

The second reason for choosing dextran instead of a mannose-based polymer is that various molecular weights of dextran polymers are commercially available, but a mannose-based polymer is not. We intended to control the extent of the *multivalent hydrogen bonding* of a polymer with TMD nanosheets by varying its molecular weight, and investigated the effect of the hydrogen bonding multivalency on the exfoliation and functionalization efficiency of TMD nanosheets in an aqueous solution. As shown in Figure 1 and Supplementary Figure 8, as the molecular weight of dextran decreased, the exfoliation efficiency of the TMD nanosheets also decreased because its hydrogen bonding multivalency was diminished.

The third reason why to choose dextran as an exfoliating and functionalizing polymer for TMD nanosheets with a selective recognition of *E. coli* O157:H7 is that it is much cheaper than a mannose-based polymer, which is much more beneficial in terms of commercialization of dex-TMDs. Hence, this dex-TMDs-based biosensor is a very cost-effective and facile method for the selective detection of pathogenic bacteria without the use of any expensive antibodies and peptides.

2. In response to 2. the Author has showed that the Raman signal for the dex-WS₂ E. coli O157:H7 is much stronger than the one observed for the E. coli K12 and E. coli O1:K1:H7. It is unclear and not sufficiently explained by the Author on which basis there is a preferential detection of E. coli O157:H7 opposed to the E. coli K12 and O1:K1:H7 strains, since they all have the same FimH protein. I would be very careful in claiming specificity of detection towards a particular strain since in literature there is still a wide debate on the binding mechanisms involved in the detection process.

I'd also ask the Author to review the use of the terms "specific" and "selective", since they have two very different meaning in analytical chemistry but the Author is using them indistinctively (see page 5 line 93 and line 96; caption in supplementary figure 12, etc.). According to the IUPAC definition (Pure Appl. Chem., Vol. 73, No. 8, pp. 1381–1386, 2001) "A specific reaction or test is one that occurs only with the substance of interest, while a selective reaction or test is one that can occur with other substances but exhibits a degree of preference for the substance of interest." The Author has not demonstrated specificity with dex-WS₂ system, since the Author has not showed comparison with other glycan-based system for E. coli detection (for example mannose). Therefore, I would suggest replacing the term with something more appropriate such as "increased binding affinity" towards a particular strain.

Response:

As discussed in the manuscript, dex-WS₂ exhibited a higher binding affinity to *E. coli* O157:H7 as compared to different strains of *E. coli* bearing a slightly different component of *fimH* lectins such as *E. coli* K-12 and *E. coli* O1:K1:H7. We speculate that the higher binding affinity of dex-TMDs to *E. coli* O157:H7 might arise from a slight structural difference in the *fimH* of *E. coli* or the recognition capability of the created 3D structure of dextran on the rigid surface of the TMD nanosheets. We are now working on finding out the 3D structure of dextran with various functional groups on the TMD surface and its docking with targets using computational simulations, which takes very long time to achieve promising results. Therefore, understanding why such selectivity to *E. coli* O157:H7 has been observed will be more investigated and the simulation-based results can be reported in the follow-up paper.

As the reviewer pointed out, we have revised the term "specificity" as "selectivity and an increased binding affinity" in the revised manuscript as shown below.

On page 15:

“To investigate further the selectivity of dex-WS₂ to *E. coli* O157:H7, other strains of *E. coli* such as *E. coli* K-12 and *E. coli* O1:K1:H7 were incubated with dex-WS₂, and then the 2LA(M) Raman signals at 352 cm⁻¹ were collected from the bacteria on glass microarrays. As shown in Supplementary Fig. 12, the strong Raman signals of dex-WS₂ were observed only from *E. coli* O157:H7. The very weak Raman signals of dex-WS₂ were measured from both *E. coli* K-12 and *E. coli* O1:K1:H7. This result clearly reveals that dex-WS₂ was able to selectively recognize *E. coli* O157:H7 against other strains of *E. coli*. It is worth noting that dex-WS₂ could differentiate *E. coli* O157:H7 from other strains of *E. coli* K-12 and *E. coli* O1:K1:H7 although they all have *fimH* on their membrane.^{48, 49} We speculate that the increased binding affinity of dex-WS₂ to *E. coli* O157:H7 might be ascribed to the fact for the slight structural difference of the *fimH* on the *E. coli* or the recognition capability of a three-dimensional structure of dextran created on the rigid surface of WS₂ nanosheets.”

3. In response to 3. Please revise your use of the terms “specific” and “selective” they are not interchangeable. Although the report is showing an effective exfoliation and functionalisation of TMDs with dextran via multiple hydrogen bonding, this contribution has provided insufficient insights and inadequate rationale and experimental materials for what they claim to be a novel and significant report in relation to Biosensing.

Response:

As the reviewer pointed out, we have revised the term “specificity” as “selectivity and an increased binding affinity” in the revised manuscript as shown above.

As we have explained above, we have the clear reasons why to choose dextran as an exfoliating and functionalizing molecule for the selective detection of pathogenic bacteria instead of a mannose-based polymer. From the Biosensing point of view, we hypothesized that if TMDs would be functionalized with multivalent dextran (MW 40,000), an enhancement effect on the binding affinity of resulting dex-TMDs to *fimH*-containing bacteria, arising from a cluster glycoside effect and a created 3D structure of dextran on the rigid surface of TMDs, would be more distinctly observed as compared to glucose because glucose has a lower intrinsic affinity to *fimH* lectin. Then, we thought that the enhanced binding affinity of dex-TMDs would eventually enable the selective detection of the pathogenic bacteria without the

use of expensive antibodies. As we anticipated, the TMD nanosheets functionalized by dextran (dex-TMDs) exhibited a significantly-enhanced binding affinity to *E. coli* O157:H7, allowing the selective detection of *E. coli* against *Salmonella typhimurium* and *Staphylococcus* (*S.*) *aureus* (Figure 5 and 6). In addition, dex-TMDs exhibited a much higher binding affinity to *E. coli* O157:H7 as compared to different strains of *E. coli* bearing a slightly different component of *fimH* lectins such as *E. coli* K-12 and *E. coli* O1:K1:H7.

In terms of biosensing, the dex-TMDs-based biosensor has clear novelty and scientific significance. It is the first report to impart a selective recognition capability to TMD nanosheets using a polymer during an exfoliation process. As stated in the manuscript, the approach based on *multivalent hydrogen bonding* enabled the simultaneous exfoliation and functionalization of TMD nanosheets in an aqueous solution. The developed dex-TMDs were then able to selectively and sensitively detect *E. coli* O157:H7 without the use of any expensive antibodies and peptides, which is clearly novel and a significant advance in the field of biosensing. In addition, the dex-TMDs-based biosensor allows the rapid, facile, and cost-effective detection of pathogenic bacteria. The dex-TMDs-based bacteria detection from sample preparation to measurement can be completed within 2 h. As any expensive antibodies and peptides are not employed, this dex-TMDs-based biosensor is very cost-effective for the selective detection of pathogenic bacteria. The dex-TMDs could detect a single copy of *E. coli* based on their Raman signal as well.

We believe that the reviewer can find the novelty and scientific significance of our work in the field of biosensing.

4. In response to 5. The Author is comparing the affinity of *E. coli* O157:H7-specific antibody found in literature with that of dex-WS₂, although the techniques used for the calibration curves are different: in the literature is used SPR and the Author is using QCM (as response to the issue raised in 5). The Author has compared the affinity of dex-WS₂ with the *E. coli* O157:H7-specific antibody against the bacteria *E. coli* O157:H7 using QCM transducer platform, which could be used as comparative technique to SPR if the exact conditions are met such as surface preparation and functionalisation process. In this case the surface functionalisation of the chip for the QCM is different from the one used in the literature, which could give a substantially different outcome.

The author has anyway compared the binding affinity of *E. coli* O157:H7-specific antibody

with dex-WS₂ for the detection of *E. coli* O157:H7, resulting in higher affinity of dex-WS₂. In the rebuttal is presented the graph of the C/ΔF vs concentration, it would be good to see also the original ΔF vs concentration. Also, there is no mention of controls experiment, with buffer for example, to have a measure of the signal to noise ratio, or with BSA for a-specific binding control. It would be useful for determining the analytical figures of merit of the sensing platform to evaluate sensitivity and limit of detection.

Response:

As the reviewer suggested, we have added the plot for the original ΔF vs concentration in Supplementary Fig. 13 as shown below. In Supplementary Fig. 13a, the frequency change for PBS was already subtracted from the one for dex-WS₂ or the antibody. When PBS flowed into the QCM chip bearing *E. coli*, the frequency change was observed to be near 1.

Based on our considerable expertise and many experiences in nanomaterials-based biosensing, BSA is sometimes helpful to prevent non-specific binding, but it also causes considerable non-specific binding. Thereby, BSA treatment was not employed on the surface modification of the QCM chip. Instead, mixed self-assembled monolayer (SAM) using 11-mercaptopundecanoic acid (MUA) and 11-hydroxy-1-undecanethiol (HUT) was employed to modify the QCM chip. These compounds can make dense SAM on the chip, in which the immobilization of a ligand and bacteria, and assays have been carried out pretty well.

We have added the experimental details for affinity measurement using QCM in the revised Supplementary Information as shown below.

Supplementary Figure 13. Measurement of the binding affinity of dex-WS₂ and an *E. coli* O157:H7-specific antibody against *E. coli* O157:H7 using a quartz crystal microbalance (QCM). a) Frequency change as a function of the concentration of dex-WS₂ or an *E. coli* O157:H7-specific antibody. b) Langmuir isotherm for the binding of dex-WS₂ or an *E. coli* O157:H7-specific antibody to *E. coli* O157:H7. All error bars represent a standard deviation from the mean values (n = 4).

On page S4:

“The affinity of dex-WS₂ and an *E. coli* O157:H7-specific antibody against *E. coli* O157:H7 using a quartz crystal microbalance (QCM) instrument was measured. After self-assembling 11-mercaptopundecanoic acid (MUA, 10 mM) and 11-hydroxy-1-undecanethiol (HUT, 50 mM) for 8 h at 40 °C on a gold QCM chip, the carboxyl group of MUA on the chip was activated by EDC/NHS (10 mM EDC and 20 mM NHS) for 1 h. Then, an antimicrobial peptide (KNYSSSIHHC) (20 mL, 0.1 mg mL⁻¹) was added into the chip and reacted for 24 h to immobilize *E. coli* O157:H7 on the QCM chip. After washing the chip with DI water and EtOH, the bacteria solution (5×10^6 CFU mL⁻¹) flowed into the chip at a rate of 50 μL min⁻¹ for 30 min. After washing it with PBS for 30 min, dex-WS₂ or the antibody flowed into the QCM chip by changing its concentration from 2 to 16 nM for 30 min. Finally, the chip was washed with PBS for 30 min.”

REVIEWERS' COMMENTS:

Reviewer #4 (Remarks to the Author):

The Author has addressed all the other comments except for the following one.

The Author is claiming that they have clearly demonstrated through sensing experiments that dex-TMDs are providing a "selective detection of E. Coli against Salmonella typhimurium and Staphylococcus(S.) aureus(Figure 5 and 6)". Unfortunately, there is no evidence in the main body or in the Supporting Information section, of a sensing experiment designed to support a selective detection. As already pointed out in my previous comments (as per the IUPAC definition), selectivity would be demonstrated when dex-TMD detects E. coli in presence of other interferences. The sensing experiments that the author has designed demonstrated a strong binding affinity towards E. Coli compared with Salmonella typhimurium and Staphylococcus(S.) aureus, but no selective detection. My suggestion is either redesigning the sensing experiment where the dex-TMD would be exposed to a matrix containing E. coli and interferences, or making some amendment on the use of the term "selective".

Response to Reviewers' comments:

Reviewer #4 (Remarks to the Author):

1. The Author has addressed all the other comments except for the following one.

The Author is claiming that they have clearly demonstrated through sensing experiments that dex-TMDs are providing a “selective detection of *E. Coli* against *Salmonella typhimurium* and *Staphylococcus(S.) aureus*(Figure 5 and 6)”. Unfortunately, there is no evidence in the main body or in the Supporting Information section, of a sensing experiment designed to support a selective detection. As already pointed out in my previous comments (as per the IUPAC definition), selectivity would be demonstrated when dex-TMD detects *E. coli* in presence of other interferences. The sensing experiments that the author has designed demonstrated a strong binding affinity towards *E. Coli* compared with *Salmonella typhimurium* and *Staphylococcus(S.) aureus*, but no selective detection. My suggestion is either redesigning the sensing experiment where the dex-TMD would be exposed to a matrix containing *E. coli* and interferences, or making some amendment on the use of the term “selective”.

Response:

Although the binding affinity of dex-TMDs to *E. coli* O157:H7 was clearly found to be much higher than that to *Salmonella typhimurium* and *S. aureus*, the selective detection of *E. coli* O157:H7 using dex-TMDs from a mixture containing the target *E. coli* and interferences was not carried out. Therefore, as the reviewer recommended, we have revised the term “selective detection” into “effective detection”, “sensitive detection” or “detection” in the revised manuscript as shown below.

On page 2:

“The dex-TMDs can *effectively detect* a single copy of *E. coli* based on their Raman signal.”

On page 5:

“Herein, we report an effective strategy for the simultaneous exfoliation and functionalization of TMD nanosheets in an aqueous solution via multivalent hydrogen bonding of a carbohydrate polymer dextran for *the sensitive optical detection of bacteria* without the use of antibodies.”

On page 14:

“This result reveals again that dex-WS₂ was able to *recognize E. coli* without the use of antibodies.”

On page 15:

“This result clearly reveals that dex-WS₂ was able to *effectively recognize E. coli* O157:H7 against other strains of *E. coli*.”

“This lower K_d value suggests that dex-WS₂ with glucan multivalency has a higher binding affinity to *E. coli* than the antibody, aptamer, and mannose, which leads to *the sensitive recognition of the target bacteria*.”

On page 16:

“These results reveal that dex-WS₂ nanosheets with glucan multivalency were able to *effectively detect E. coli* pathogens in a single copy without the use of antibodies.”